# Systematically attenuating DNA targeting enables CRISPR-driven editing in bacteria

Daphne Collias [1,2], Elena Vialetto[1], Jiaqi Yu [1], Khoa Co[1], Éva d. H. Almási [3], Ann-Sophie Rüttiger[1], Tatjana Achmedov[1], Till Strowig [3,4] & Chase L. Beisel [1,2,5] ✉

Bacterial genome editing commonly relies on chromosomal cleavage with Cas nucleases to counter-select against unedited cells. However, editing normally requires efficient recombination and high transformation efficiencies, which are unavailable in most strains. Here, we show that systematically attenuating DNA targeting activity enables RecA-mediated repair in different bacteria, allowing chromosomal cleavage to drive genome editing. Attenuation can be achieved by altering the format or expression strength of guide (g)RNAs; using nucleases with reduced cleavage activity; or engineering attenuated gRNAs (atgRNAs) with disruptive hairpins, perturbed nuclease-binding scaffolds, non-canonical PAMs, or guide mismatches. These modifications greatly increase cell counts and even improve the efficiency of different types of edits for Cas9 and Cas12a in *Escherichia coli* and *Klebsiella oxytoca*. We further apply atgRNAs to restore ampicillin sensitivity in *Klebsiella pneumoniae*, establishing a resistance marker for genetic studies. Attenuating DNA targeting thus offers a counterintuitive means to achieve CRISPR-driven editing across bacteria.

The study and engineering of bacteria have been vastly improved with CRISPR technologies and the ongoing advances in CRISPR-based editing[1,2]. Traditionally, bacteriophage-derived DNA recombinases are combined with RNA-guided Cas nucleases to achieve efficient editing ranging from base changes to large deletions and insertions[3,4]. The recombinases drive the homologous recombination of a DNA repair template (RT), while the Cas nucleases counterselect against unedited cells through the generation of cytotoxic double-stranded DNA breaks to the unedited bacterial chromosome[3]. The flexibility of this approach allows for both small and large edits and thus remains the method of choice for editing in bacteria despite a growing set of other options[5–9]. However, CRISPR-based counterselection typically requires high transformation efficiencies and the availability of compatible phage-based recombinases unavailable in most bacteria. Even with improvements that delay counterselection with inducible DNA targeting[10–13] or limit bacterial escape with RecA inhibitors[14], CRISPR-

based counterselection still remains largely off-limits outside of model bacteria and, even in model bacteria, incredibly difficult for challenging edits such as larger insertions, multiplexed editing, and libraries.

Prior work reported an intriguing exception to CRISPR-based counterselection: chromosomal cleavage by Cas9 in *Escherichia coli* can be actively repaired through homologous recombination[15]. Recombination was hypothesized to come from weaker targeting at certain sites, which left uncleaved copies of the chromosome that could mediate homologous recombination and maintain cell viability. Recombination was dependent on RecA and, in one instance, drove integration of a plasmid-encoded RT in the absence of a heterologous recombinase. Under this setup, driving homologous recombination of a supplied RT would achieve flexible editing while circumventing the need for heterologous recombinases or high transformation efficiencies due to enhanced cell survival. However, survival was seemingly random and site-dependent, creating uncertainties about

[1]Helmholtz Institute for RNA-based Infection Research (HIRI), Helmholtz Centre for Infection Research (HZI), 97080 Würzburg, Germany. [2]Department of Chemical and Biomolecular Engineering, North Carolina State University, 27695 Raleigh, NC, USA. [3]Helmholtz Centre for Infection Research (HZI), 38124 Braunschweig, Germany. [4]German Center for Infection Research (DZIF), Partner Site Hannover-Braunschweig, Braunschweig, Germany. [5]Medical Faculty, University of Würzburg, 97080 Würzburg, Germany. ✉ e-mail: chase.beisel@helmholtz-hiri.de

whether CRISPR-driven homologous recombination could be broadly achieved in bacteria.

Here, we show that CRISPR-driven homologous recombination can be systematically achieved by attenuating DNA targeting activity, in different cases boosting the number of transformants as well as the editing efficiency. The most tunable approach, which involved what we call attenuated gRNAs (atgRNAs) named based on modifications designed to interfere with gRNA function, could achieve flexible editing with Cas9 and Cas12a nucleases in different bacteria and mediate small base exchanges and large insertions. The approach obviated the need for a heterologous recombinase and greatly increased the number of recovered colonies without sacrificing editing efficiency, the two major drawbacks of traditional recombination-based editing in bacteria. The use of atgRNAs thus represents a paradigm for CRISPR-based editing in bacteria that achieves improved editing outcomes by scaling back targeting activity.

## Results

### Altering gRNA format and expression can boost cell counts and genome editing

We were initially intrigued why targeting some locations in the *E. coli* genome with the *Streptococcus pyogenes* Cas9 led to RecA-dependent homologous recombination rather than cell death[15]. One observation from this work was that the tested gRNAs were all encoded as CRISPR arrays, with each array expressed from its native promoter from *S. pyogenes*. The transcribed arrays are processed into CRISPR RNAs (crRNAs) through the action of a separate trans-activating crRNA (tracrRNA)[16]. In contrast to the CRISPR arrays, single-guide RNAs (sgRNAs), which circumvent the need for a separate tracrRNA, are commonly used for Cas9-based editing in bacteria[17]. Each sgRNA is also normally expressed from a heterologous promoter[11,14,18,19]. We refer to the processed crRNA:tracrRNA duplex and the sgRNA as gRNAs. We thus asked if this difference in gRNA format or expression accounts for

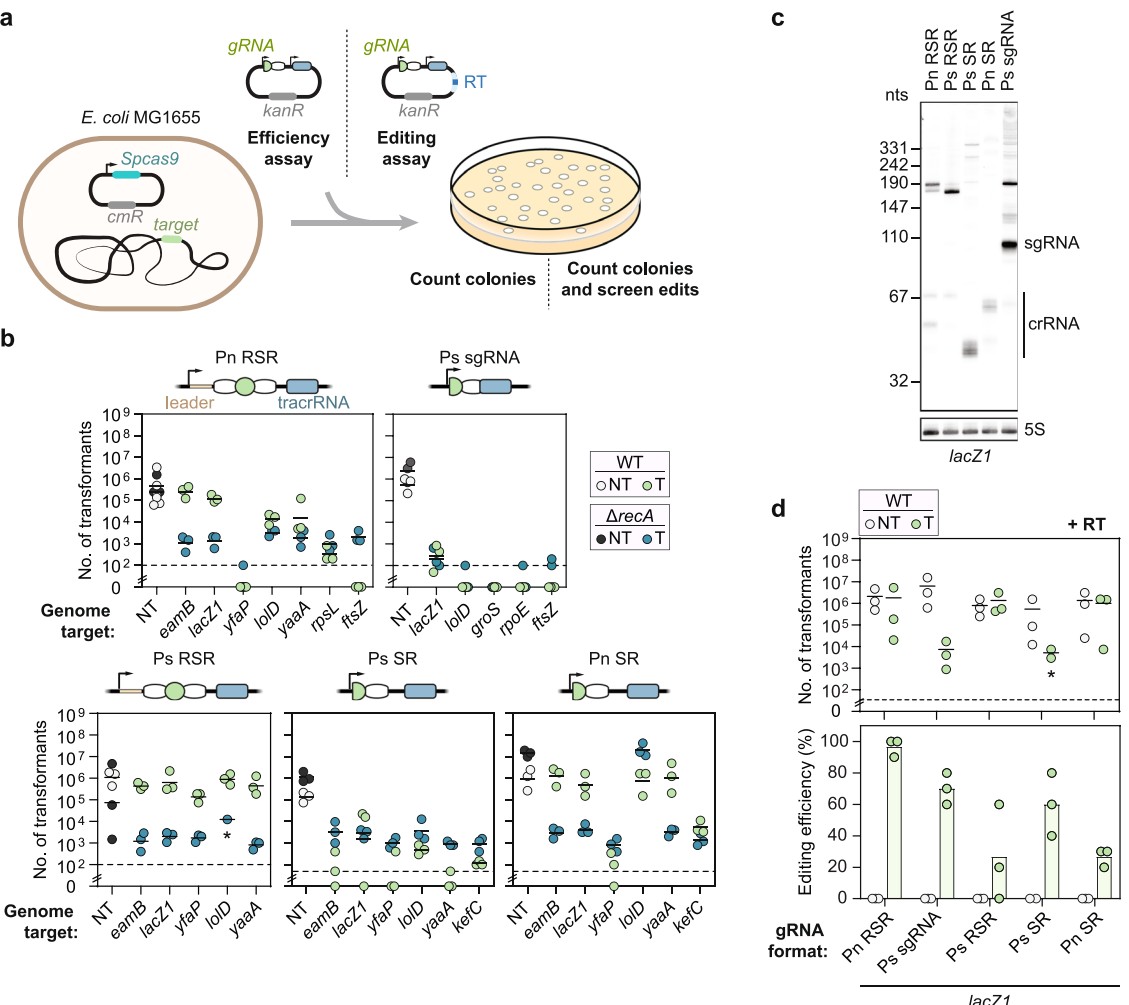

**Fig. 1 | The outcome of chromosomal targeting in *E. coli* depends on gRNA format and abundance. a** Schematic of the experimental setup for chromosomal targeting and editing. **b** Genome targeting assay in *E. coli* and *E. coli* Δ*recA*. **c** Northern blot analysis of whole RNA isolated from *E. coli* Δ*lacI-Z* with pCas9 and pgRNA. A *lacZ1* spacer specific probe was used to probe the abundance of each RNA product. A 5 S probe was used as a control on the same gel and shown below the *lacZ1* probed gel. The approximate size of an sgRNA is indicated to the left of the Northern blot and the approximate size of mature crRNAs are indicated with a line. **d** Genome editing assay in *E. coli* targeting *lacZ1* to introduce AvrII restriction enzyme recognition site as a silent mutation. Individual dots for the transformations indicate a single biological replicate. * indicates that the transformants resulted in a lawn or

uncountable colonies. Dashed lines indicate the limit of detection from plating. NT and T indicate targeting and non-targeting gRNAs, respectively. Individual dots for the editing efficiencies indicate the average of 3 colonies screened from one biological replicate for NT samples or 10 colonies screened from one biological replicate for targeting samples. Bars indicate the mean of the dots. The dashed line in (**b, d**) indicates the limit-of-detection. The limit-of-detection was calculated based on the volume of cells plated for each experiment. The mean number of transformants (indicated by a horizontal line) was not calculated for samples with biological replicates that fell below the limit of detection. WT indicates wild-type *E. coli* MG1655, Δ*recA* indicates *E. coli* MG1655 Δ*recA*, NT indicates non-targeting, T indicates targeting, and RT indicates repair template.

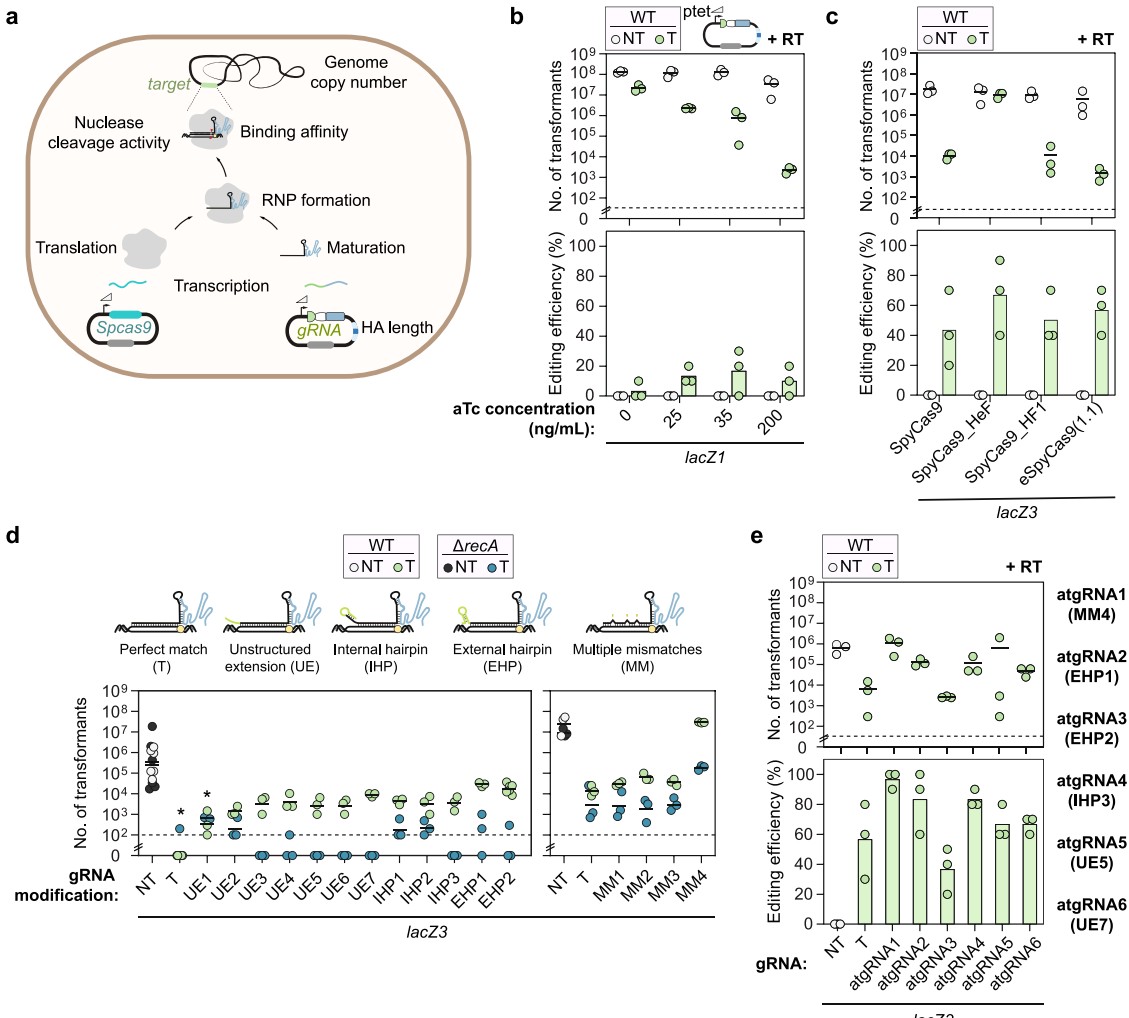

**Fig. 2 | Modulating DNA targeting activity, including through attenuated gRNAs, can boost colony counts without sacrificing editing efficiencies.**
**a** Schematic of different steps that can be altered to attenuate DNA targeting to improve homologous recombination with a supplied RT. The maturation step applies only to crRNAs. **b** Genome editing assay in *E. coli* with aTc inducible sgRNA expression. A non-selective out-growth with aTc induction was used prior to selective plating. **c** Genome editing assay in *E. coli* using WT, HeF, HF1, and e(1.1) SpyCas9. **d)** Genome targeting assay in *E. coli* and *E. coli* Δ*recA* with modified sgRNAs and Cas9. **e** Genome editing assay in *E. coli* using selected attenuated gRNAs (atgRNAs) from d. Cells were made electrocompetent at ABS₆₀₀ ≈ 1.4–1.6 for the editing assay. Individual dots for the transformations indicate a single biological replicate. Dashed lines indicate the limit of detection from plating. NT indicates non-targeting and T indicates targeting. Individual dots for the editing efficiencies (in **b**, **c**, **e**) indicate the average of 3 colonies screened from one biological replicate for NT samples or 10 colonies screened from one biological replicate for targeting samples. Bars indicate the mean of the dots. * indicates that the transformants resulted in a lawn or uncountable colonies. The dashed line in **b**–**e** indicates the limit-of-detection. See Fig. 1 for details. The mean number of transformants (indicated by a horizontal line) was not calculated for samples with biological replicates that fell below the limit of detection. WT indicates wild-type *E. coli* MG1655, Δ*recA* indicates *E. coli* MG1655 Δ*recA*, NT indicates non-targeting, T indicates targeting, and RT indicates repair template.

RecA-mediated cell survival, helping us work toward CRISPR-driven editing in bacteria.

Using a standard plasmid transformation assay in *E. coli* in which a gRNA plasmid is transformed into cells already harboring a Cas9 plasmid (Fig. 1a), we targeted genomic sites that previously yielded different extents of cell survival in the presence of *recA*[15]. Under this setup, cell counts of the WT and Δ*recA* strains are expected to depend on targeting activity. Namely, strong targeting activity would outpace RecA-mediated repair, yielding low cell counts with or without *recA*. In contrast, weak targeting activity would allow RecA-mediated repair, resulting in higher cell counts in the WT strain compared to the Δ*recA* strain. Finally, limited targeting activity would require little to no repair, resulting in high cell counts in both the WT and Δ*recA* strains. Similar to the prior work[15], applying the transformation assay using minimal CRISPR arrays containing a spacer flanked by full-length repeats (RSR) and driven by the native promoter (Pn RSR) yielded

varying extents of survival in the presence of *recA* (0 - ~10⁵ transformants) and consistently low survival in the absence of *recA* (~10³ transformants) compared to a non-targeting (NT) control (~10⁵ transformants) (Fig. 1b). In contrast, sgRNAs driven by a constitutive synthetic promoter (Ps sgRNA) consistently yielded virtually no cell counts in the presence or absence of *recA* for the same genomic target sites (Fig. 1b and Supplementary Fig. 1a). Target locations alone therefore cannot explain RecA-mediated cell survival and instead point to factors related to gRNA format and expression.

To tease apart gRNA format and expression, we replaced the native promoter in front of the minimal CRISPR array with a constitutive synthetic promoter (Ps RSR) (Fig. 1b). We also expressed a semi-processed crRNA containing a 20-bp spacer followed by a full-length repeat (SR) with either promoter (Ps SR, Pn SR) to interrogate any contributions from processing. Combining the synthetic promoter with the native array (Ps RSR) or the native promoter with the

semi-processed crRNA (Pn SR) generally yielded elevated colony counts in the WT strain (9/11 targets with ~$10^5$ transformants), while combining the synthetic promoter with the semi-processed crRNA (Ps SR) yielded low colony counts (5/5 targets with 0 - ~$10^3$ transformants). Colony counts were consistently low in the absence of *recA* (15/17 targets with ~$10^3$ transformants), underscoring the importance of RecA-mediated homologous recombination. These data suggest that changing the format and expression of the gRNA can alter the outcome of survival.

The operating hypothesis is that weaker targeting allows cells to survive through RecA-mediated recombination with replicated copies of the genome[15]. Following this hypothesis and the observed impact of colony counts on gRNA format, we would expect weaker targeting to derive from lower gRNA abundance that could direct fewer Cas9 molecules to cut target DNA, allowing RecA-mediated repair to out-compete extensive DNA cutting. In line with our expectation, northern blotting analysis on the complete set of gRNAs targeting *lacZ1* revealed that lower final gRNA abundances were tied to improved survival in the presence of *recA* (Fig. 1c and Supplementary Fig. 1b). The crRNA sizes also varied, possibly due to incomplete crRNA processing in *E. coli* as well as differences in transcriptional start sites between promoters; the size variations could be contributing to the extent of DNA targeting. Similar trends in gRNA size and abundance were observed targeting two other locations in *lacZ* (Supplementary Fig. 1b, c). To explicitly evaluate the impact of expression strength, we replaced the strong constitutive promoter in front of the *lacZ1*- and *lolD*-targeting sgRNAs with two weaker versions: a medium promoter (J23116) and a weak promoter (J23109)[20]. The weak promoter yielded similarly high colony counts (~$10^5 - 10^6$ transformants) for WT and Δ*recA* cells for both sites, indicative of cell survival. In contrast, the medium promoter yielded low colony counts (0 - ~$10^2$ transformants) in WT and Δ*recA* strains similar to the strong promoter, indicative of cell death (Supplementary Fig. 1d). In total, these results provide direct evidence that reduced gRNA abundance can lead to cell survival through RecA-mediated recombination in the absence of a provided RT.

If the cells survive genome targeting by Cas9 via RecA-mediated recombination, then the presence of a RT whose edit prevents targeting should lead to genome editing without sacrificing colony counts. We therefore introduced a plasmid-encoded RT to mutate part of the *lacZ1* target into a restriction site, and we assessed colony counts and the editing efficiency (Fig. 1d and Supplementary Fig. 1e). All gRNA formats yielded variable colony counts (~$10^4 - 10^6$ transformants) that paralleled those in the absence of the RT, with Ps sgRNA and Ps SR yielding the fewest colonies (~$10^4$ transformants) and the other formats (Pn RSR, Ps RSR, Pn SR) yielding the most colonies (~$10^5 - 10^6$ transformants). Intriguingly, Pn RSR yielded the highest editing efficiency (~97%) even above that by the sgRNA driven by a constitutive synthetic promoter (Ps sgRNA) (70%), while the other formats yielded more variable yet measurable editing (~20–60%) (Fig. 1d). In total, changing gRNA format and expression can achieve CRISPR-driven editing that boosts colony counts, obviates the need for an exogenous recombinase, and can preserve the efficiency of precise editing.

### Systematically attenuating genome targeting with Cas9 can increase colony counts and editing efficiencies

The impact of gRNA format and expression suggested that purposefully attenuating DNA targeting activity could not only increase colony counts but also even improve the editing efficiency. If the goal is to attenuate targeting, we can envision multiple means that weaken or delay any step spanning gRNA and Cas9 production to DNA cleavage (Fig. 2a), including altering the expression or stability of the gRNA or Cas nuclease, slowing nuclease:gRNA complex formation, and reducing DNA target recognition or cleavage. Furthermore, we reasoned that any approach could be implemented individually or together to fine-tune the activity reduction. We therefore tested various attenuation approaches beyond modifying gRNA format to determine if any of these approaches could predictably boost cell counts and editing.

We first tested the impact of modulating gRNA expression using a tetracycline-inducible promoter (Fig. 2b). This approach offered excellent flexibility, as different concentrations of the inducer anhydrotetracycline (aTc) could be readily added to tune targeting activity. We further encoded an sgRNA, as encoding either a crRNA or tracrRNA could cause the other RNA to become limiting. Accordingly, titrating the aTc concentration in cultures transformed with the sgRNA, Cas9, and RT plasmids resulted in colony count reductions varying between ~6-fold (0 ng/ml aTc) and ~$10^5$-fold (200 ng/ml aTc) compared to the NT control. When introducing mutations into the *lacZ1* target to create a restriction site, consistent editing was observed even with lower aTc concentrations without sacrificing colony counts. However, the extent of editing never exceeded an average of 20% even for the highest applied aTc concentration.

Next, we tested the impact of reducing DNA targeting activity with high-fidelity Cas9 nucleases (Fig. 2c). These nucleases were evolved to more readily reject mismatches and therefore reduce off-target DNA cleavage[21–23]. However, they also tended to exhibit lower cleavage efficiencies at otherwise perfect targets, which we hypothesized could promote RecA-mediated survival upon genome targeting even with an sgRNA conferring robust DNA targeting. We therefore tested three established high-fidelity versions of Cas9 (SpyCas9_HeF, SpyCas9_HF1, eSpyCas9(1.1)) along with the sgRNA driven by a constitutive synthetic promoter (Ps sgRNA) that resulted in extensive cell killing (Fig. 2c). We found that SpyCas9_HeF, which contains the combined mutations of both eSpCas9(1.1) and SpyCas9_HF1[22,24], exhibited much higher colony counts than the parental Cas9 (SpyCas9) and the other two variants (~$10^3$-fold), while all variants exhibited similar average editing efficiencies (~50–60%) compared to the parental Cas9 (~43%)[25,26]. In the case of SpyCas9_HeF, the colonies had a mixed genotype as determined from the restriction enzyme digests that had only faint digestion bands, suggesting that cells were still undergoing repair. Less-active nucleases, such as some high-fidelity nucleases, therefore offer a distinct means of achieving CRISPR-driven editing, although some nucleases outperform others.

As a final approach, we reasoned that modifying the gRNA itself to reduce Cas9 binding, target recognition, or target cleavage could be applied to achieve CRISPR-driven editing. Fortunately, numerous modifications are known that can have a minor to massive impact on DNA targeting, including introducing single mismatches (SM) or multiple mismatches (MM) into the guide sequence at different locations[26–30], extending the PAM-distal end of the gRNA to include an unstructured extension (UE) or a hairpin that does (IHP) or does not (EHP) extend into the guide sequence[31,32], targeting genomic sequences adjacent to non-canonical PAMs (NC)[3,33–36], and mutating the fixed region of the gRNA bound by Cas9 to affect RNA folding or Cas9 recognition[37] (Fig. 2d and Supplementary Fig. 1g). The proceeding number specifies the order in which the mutation was tested (e.g., MM3 for the third tested multiple-mismatch modification).

Using the *lacZ3*-targeting sgRNA as the starting point, the modifications gave varying degrees of colony counts in the presence or absence of *recA* (0 - ~$10^7$ transformants). For most of the modifications though, at least one variant yielded elevated colony counts in the presence versus absence of *recA* (e.g., ~$10^2$-fold increase for MM4), indicating attenuated targeting activity. Combining a subset of these attenuated gRNAs (atgRNAs) with the RT to introduce a restriction site in *lacZ* led to similar or even higher editing efficiencies compared to the original sgRNA but with modestly to greatly increased colony counts (Fig. 2e). For instance, one approach in which three target mismatches were introduced into the sgRNA guide (atgRNA1) yielded colony counts indistinguishable from a non-targeting sgRNA along with ~97% editing. Separately, atgRNA1 and atgRNA6 guiding the high fidelity eSpCas9(1.1) increased colony counts by ~$10^3$-fold and yielded

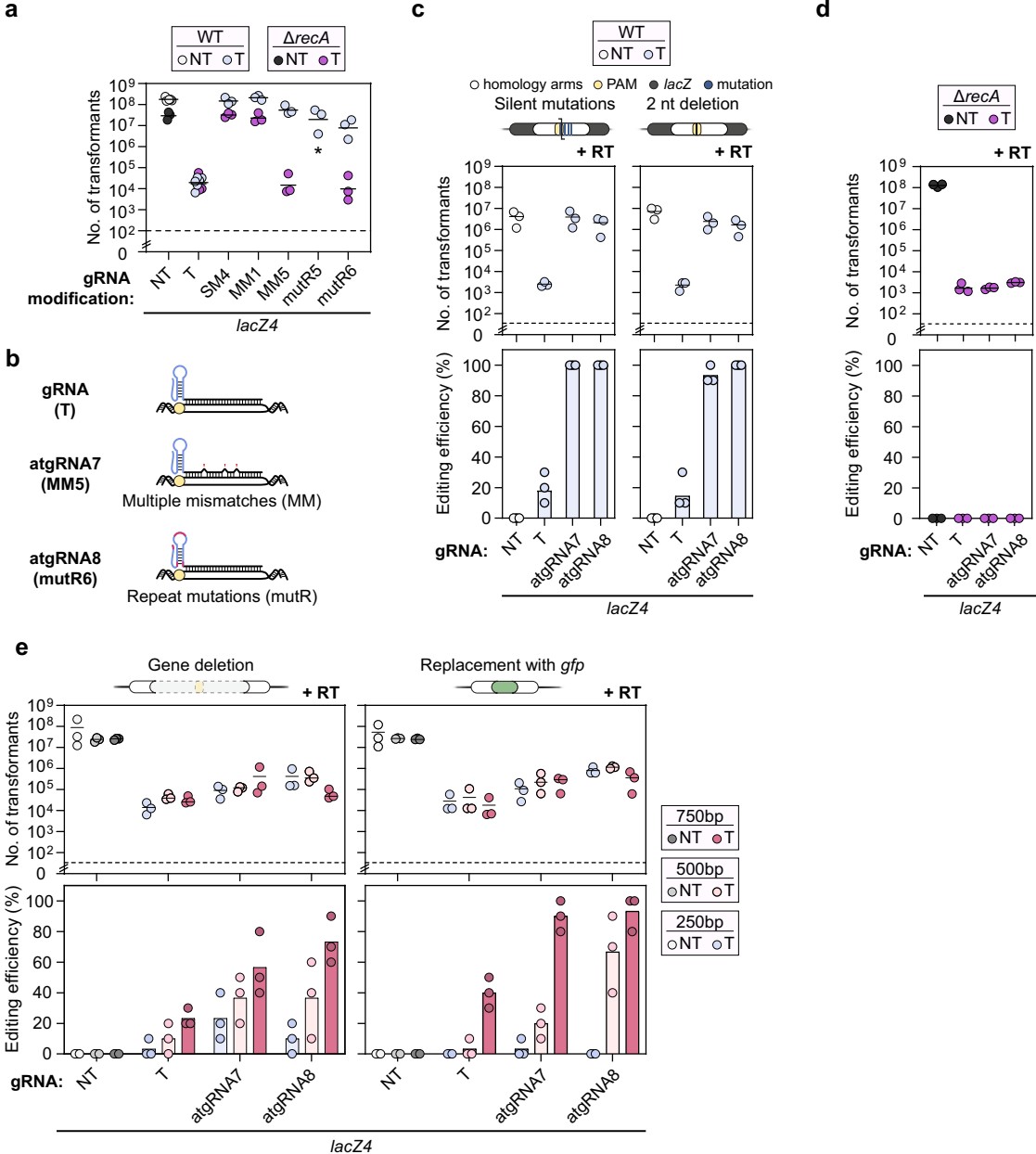

**Fig. 3 | CRISPR-driven editing with attenuated gRNAs extends to Cas12a.**
**a** Genome targeting assay in *E. coli* and *E. coli* Δ*recA* with modified gRNAs and Cas12a. **b** Schematic of atgRNAs for Cas12a. **c)** Genome editing assay in *E. coli* using atgRNAs to introduce a RE silent mutation (left panels) and a 2nt deletion in the *lacZ* ORF (right panels). **d** Editing assay to an AclI restriction enzyme site into *lacZ* in the *E. coli* Δ*recA* strain. Each dot in the editing efficiency graph represents the value obtained for 3 colonies (*n* = 3). **e** Genome editing assay in *E. coli* using atgRNAs to delete the entire *lacZ* gene (left panels) and substitute *lacZ* with *gfp* (right panels). Individual dots for the transformations indicate a single biological

replicate. NT indicates non-targeting and T indicates targeting. Individual dots for the editing efficiencies (in **c**–**e**) indicate the average of 3 colonies screened from one biological replicate for NT samples or 10 colonies screened from one biological replicate for targeting samples. Bars indicate the mean of the dots. * indicates that the transformants resulted in a lawn or uncountable colonies. The dashed line in (**a**, **c**, **d**) indicates the limit-of-detection. See Fig. 1 for details. WT indicates wildtype *E. coli* MG1655, Δ*recA* indicates *E. coli* MG1655 Δ*recA*, NT indicates non-targeting, T indicates targeting, and RT indicates repair template.

editing efficiencies of 100% and 50%, respectively (Supplementary Fig. 1f). The colonies screened for atgRNA4 and atgRNA6 resulted in only faint digestion patterns (Supplementary Fig. 1f), indicating that colonies exhibit a mixed genotype and may still be undergoing editing. In many of these cases, introducing the RT into the gRNA plasmid affected the number of transformants, although the ratio of colonies between non-targeting and targeting conditions were similar with or without the RT. In total, sufficiently attenuating DNA targeting by Cas9 through different means, either individually or in combination, can

increase colony counts while preserving the editing efficiency via RecA-mediated homologous recombination.

### CRISPR-driven editing with atgRNAs is generalizable beyond Cas9

While Cas9 is arguably the most widely used nuclease for genome editing in bacteria, a growing suite of DNA-targeting Cas nucleases are becoming available[38] whose targeting activity could be attenuated to enhance genome editing in bacteria. One increasingly popular

DNA-targeting nuclease is Cas12a[39]. Apart from being structurally and functionally distinct from Cas9, Cas12a can process a transcribed CRISPR array without any accessory factors[39] and recognizes T-rich PAMs[39] distinct from those recognized by Cas9[40]. Cas12a was also employed in one bacterial application when Cas9 proved cytotoxic[41]. We therefore asked if atgRNAs specific to Cas12a could be generated to achieve CRISPR-driven editing in bacteria.

We focused on the widely-used Cas12a from *Acidaminococcus* species (AsCas12a)[39] given its widespread use for CRISPR technologies. After confirming that designed gRNAs consistently lead to low cell counts (~$10^2$–$10^3$ transformants) when targeting the *E. coli* genome in the absence of *recA* (Supplementary Fig. 2a), we generated variants of a *lacZ*-targeting gRNA in line with the Cas9 atgRNAs. Specifically, we appended an external hairpin on the PAM-distal end of the guide (EHP), introduced a mutation in the Cas12a repeat shown to partially inhibit DNA targeting[42] (mutR), mutated the leader region upstream of the Cas12a repeat to introduce a hairpin that can also inhibit DNA targeting[43] (IRH), targeted sites with non-canonical PAMs (NC)[40], and introduced a single target mismatch (SM) or multiple target mismatches (MM)[27] in the guide sequence (Fig. 3a and Supplementary Fig. 2b, c). As with Cas9, at least one variant for most of the tested modifications yielded elevated colony counts in the presence versus absence of *recA* (~$10^4$-fold for MM5). Using two of these variants (MM5, mutR6) (Fig. 3b), we assessed genome editing by introducing either a silent mutation that generates a restriction site or a two-nucleotide deletion at the *lacZ* target site (Fig. 3c and Supplementary Fig. 2d, e). In both cases, the atgRNAs yielded ~1000-fold greater colony counts compared to the original gRNAs. Critically, the editing efficiencies were also much higher for the atgRNAs (95–100%) versus the original gRNAs (10–30%). High colony counts and editing was lost in the absence of *recA* (Fig. 3d), confirming the importance of RecA-mediated homologous recombination with atgRNAs. Therefore, CRISPR-driven editing in bacteria can be achieved with atgRNAs that extend beyond Cas9.

Given the superior performance of Cas12a atgRNAs over gRNAs when generating small edits, we asked how they perform when generating larger edits. We attempted two different types of large edits: the deletion of *lacZ* (~3.1 kb) and the replacement of *lacZ* with a 717-bp fragment encoding *gfp* (Fig. 3e and Supplementary Fig. 2f). As these types of edits are typically more difficult to create, we varied the length of the homology arms of the RT to help improve the efficiency of homologous recombination[44–46]. While the atgRNAs did not fully recover colony counts for these large edits, they still yielded higher colony counts than the original gRNA (e.g., 13-fold for atgRNA7 and 20-fold for atgRNA8 for *gfp* replacement with 750-bp homology arms). The atgRNAs also yielded higher editing efficiencies than the original gRNA (e.g., ~10% vs. ~40% for gene deletion with 500-bp homology arms), with longer homology arms consistently increasing the editing efficiency (e.g., ~10% with 250-bp homology arms vs. ~70% with 750-bp homology arms for gene deletion with atgRNA8). Therefore, when testing more challenging edits, atgRNAs can outperform the original gRNA in both colony counts and editing efficiencies.

**atgRNAs can improve editing performance in *Klebsiella* species**

Building on the demonstrations of CRISPR-driven editing in *E. coli*, we asked how well this approach could extend to bacteria with less developed tools. As a first demonstration, we focused on *Klebsiella oxytoca*, a commensal bacterial species recently shown to reduce the intestinal colonization of multidrug-resistant (MDR) *Klebsiella pneumoniae* in various mouse models[47]. While Cas9-based editing with the λ-*red* recombination system had been implemented in one strain of this species[48], this approach required extensive sgRNA screening and generally resulted in few transformants. Applying atgRNAs therefore offered an opportunity to expand the editing tools in this clinically relevant species while also setting the stage to further interrogate the mechanistic basis of colonization resistance.

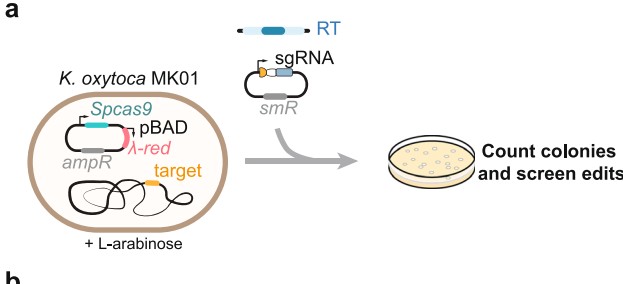

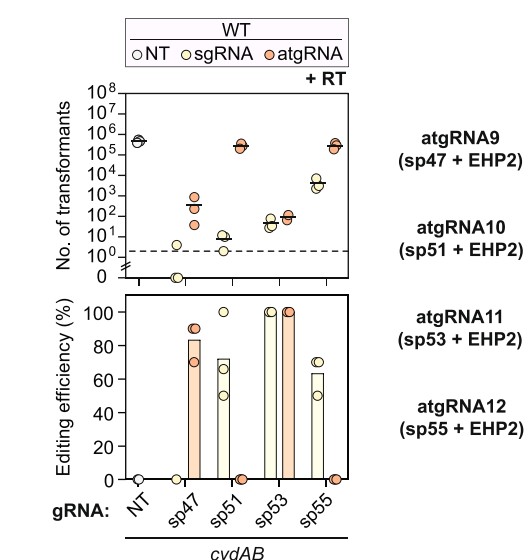

**Fig. 4 | An attenuated gRNA enhances editing in *Klebsiella oxytoca*. a** Schematic of genome editing experimental setup. **b** Genome targeting assay in *K. oxytoca* with sgRNAs and atgRNAs to delete *cydAB*. Individual dots for the transformations indicate a single biological replicate. Dashed lines indicate the limit of detection from plating. NT indicates non-targeting. Individual dots for the editing efficiencies indicate the average of 3 colonies screened from one biological replicate for NT samples or 10 colonies screened from one biological replicate for targeting samples. Bars indicate the mean of the dots. The dashed line indicates the limit-of-detection. See Fig. 1 for details. The mean number of transformants (indicated by a horizontal line) was not calculated for samples with biological replicates that fell below the limit of detection. WT indicates wild-type *K. oxytoca* MK01, NT indicates non-targeting, and RT indicates repair template.

For the test case within the *K. oxytoca* strain MK01, we sought to delete a 2.7-kb fragment of the *cydAB* operon previously shown to be important for growth under microaerobic conditions in *E. coli*[49]. We began with the Cas9 editing system previously established in this species[48] (Fig. 4a) to provide a direct basis of comparison and because the absence of λ-*red* did not yield any detectable edits with a linear or plasmid RT (Supplementary Fig. 3b, c). The necessity of λ-*red* deviated from what we observed in *E. coli*, although this may be attributed to the different strain, the use of a linear repair template, or the larger deletion at this particular genomic site.

While the four tested sgRNAs yielded different extents of colony counts, one sgRNA (sp47) yielded the lowest colony counts (~1 transformant) and no detectable edits (Fig. 4b). When considering which sgRNA modifications to introduce, we opted to introduce the hairpin EHP2 to the PAM-distal end of each sgRNA because it can be introduced independently of the target sequence (Fig. 4b and Supplementary Fig. 3d). For the sp47 sgRNA, the resulting atgRNA boosted colony counts to be consistently above the limit of detection (~$10^2$ transformants) and yielded an editing frequency of ~83% (Fig. 4b). For the other sgRNAs, addition of the external hairpin maintained colony counts and high editing frequencies (sp53) or greatly boosted the colony counts

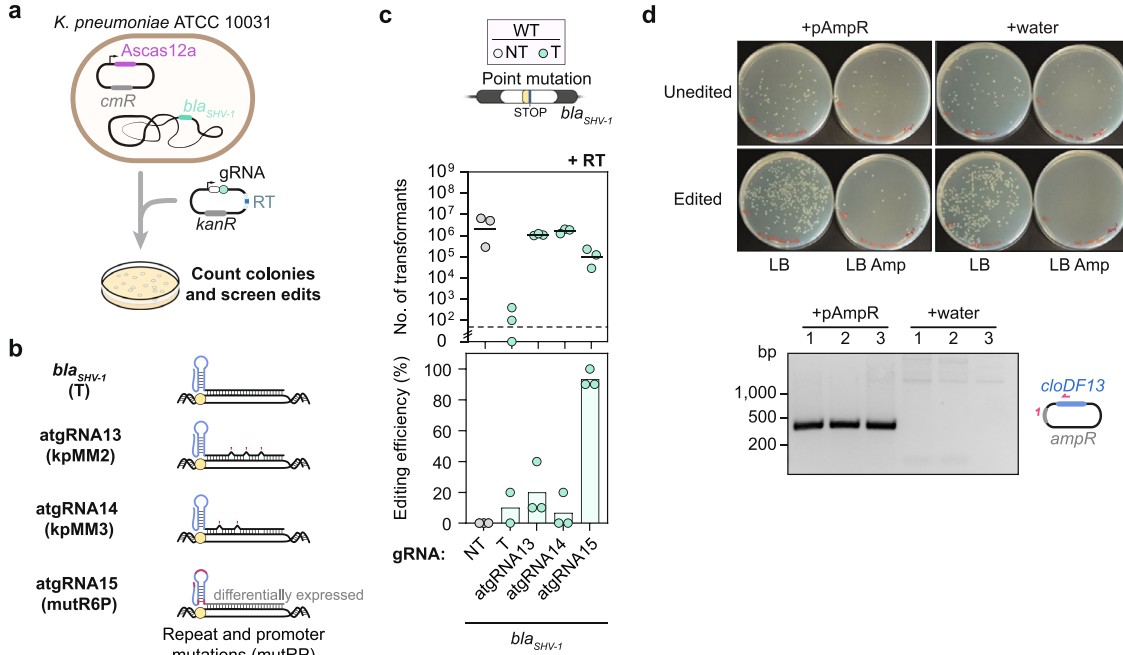

**Fig. 5 | Attenuated gRNAs can be applied to reverse antibiotic resistance in multidrug-resistant Klebsiella pneumoniae. a** Schematic of genome editing assay. **b** Schematic of canonical and attenuated guide design. **c** Genome editing assay in *K. pneumoniae* to introduce a stop codon in the *bla*$_{SHV-1}$ gene. Each dot in the editing plot is the result obtained after screening 10 and 3 colonies respectively for the T and NT samples. Bars indicate the mean of the dots. For the T gRNA, each dot represents the editing efficiency obtained for 4 colonies due to the absence of other colonies on the plates. One of the biological replicates yielded no colonies for this condition. The dashed line indicates the limit-of-detection. See Fig. 1 for details. The mean number of transformants (indicated by a horizontal line) was not calculated for samples with biological replicates that fell below the limit-of-detection.

**d** Top: plate images upon transformation of a single biological replicate of edited cells (atgRNA15) and unedited cells (NT) with the pAmpR plasmid or water in Amp or LB only plates. Bottom: schematic of pAmpR amplification by PCR with confirmatory gel image. The three lanes on the left are PCR amplicons from three individual colonies after plating the edited cells that were transformed with pAmpR, while the three lanes on the right are PCR amplicons from three individual colonies after plating edited cells that were transformed with water (from LB plates). The expected band size for the amplicon is ~400 bp. WT indicates wild-type *K. pneumoniae* ATCC 10031, NT indicates non-targeting, T indicates targeting, and RT indicates repair template.

but at the expense of editing (sp51, sp55). In these cases, DNA targeting was likely too weak to drive editing, at least within the timeframe of the experiment. sp55 was particularly intriguing given the higher colony counts (~10⁴ transformants) and editing frequencies (~60%) without any modification (Fig. 4b), suggesting that the sgRNA already exhibited attenuated targeting possibly through sgRNA misfolding or poor target recognition. Overall, expedient testing of one atgRNA modification enhanced editing, although more extensive screening of atgRNA modifications could further enhance editing across target sites.

Beyond *K. oxytoca*, we turned to a second demonstration of CRISPR-driven editing outside of *E. coli*: eliminating an antibiotic resistance marker in *Klebsiella pneumoniae* to facilitate new genetic tools. *K. pneumoniae* is commonly associated with antibiotic resistance, complicating treatment of infections. In addition, resistance greatly restricts which antibiotics can be applied to select for plasmids, limiting genetic studies and efforts to combat future infections. We utilized Cas12a to explore CRISPR-driven editing outside of *E. coli* utilizing a nuclease besides Cas9.

We sought to reverse resistance to ampicillin by disrupting *bla*$_{SHV-1}$, an intrinsic β-lactamase gene common to *K. pneumoniae* strains[50]. We focused on the commonly studied strain ATCC 10031 that is sensitive to chloramphenicol but not ampicillin (Fig. 5a). A Cas12a gRNA targeting *bla*$_{SHV-1}$ yielded virtually no colonies in the presence or absence of *recA*, offering a starting point to generate atgRNAs (Supplementary Fig. 3e). Of the tested atgRNAs, two harboring target mismatches in the guide (kpMM2, kpMM3) and one containing a mutated Cas12a repeat combined with a mutation in the constitutive gRNA promoter (mutR6P) led to elevated colony counts in the presence versus the absence of *recA* (e.g., ~10⁵ vs. 0 transformants) (Supplementary Fig. 3e). Proceeding with

these three atgRNAs and a RT intended to generate a premature stop codon (Fig. 5b), we found that all three yielded >1000-fold increase in colony counts compared to the original gRNA (Fig. 5c). In addition, atgRNA15 yielded ~95% editing compared to ~10% editing with the original gRNA. The other two atgRNAs yielded similarly low editing as the original gRNA, albeit with much greater colony counts that facilitates finding an edited colony.

Continuing with the strain edited with atgRNA15, we evaluated transformation of the pAmpR plasmid conferring resistance to ampicillin (Fig. 5d). When growing cells on ampicillin plates, the edited strain yielded colonies only in the presence of pAmpR, while the wildtype strain yielded colonies in the presence or absence of pAmpR. Screening edited colonies that formed with ampicillin confirmed the presence of pAmpR. The edit therefore expands the genetic toolbox available to this strain with an additional selectable marker. Overall, we conclude that atgRNAs can be applied to achieve CRISPR-driven editing in non-model bacteria even for large edits, in some cases outperforming the more traditional CRISPR-based counterselection.

## Discussion

In this work, we showed that systematically attenuating DNA targeting activity can achieve CRISPR-driven editing in bacteria, greatly boosting colony counts and even increasing the frequency of precise genome editing. This seemingly paradoxical concept—making DNA targeting weaker can improve editing—may be explained by the cells having time to repair cut genomic DNA with other copies of the chromosome or a provided RT. During exponential growth, bacteria initiate genome replication multiple times in one cell cycle to keep pace with cell division. Here, attenuated targeting would leave more genomic copies

intact and thus available for templated repair. However, other mechanisms could be at work. For instance, some atgRNA modifications could promote active release of cut DNA, allowing faster access by the repair machinery. In addition, attenuated targeting may lead to nicking rather than cleavage of the DNA target[28,51], which can also be repaired through homologous recombination[52]. Within any of these mechanisms, templated repair through genomic DNA is likely more efficient, although repair with the RT prevents retargeting and thus drives the eventual build-up of edited cells in the population. Because of cell survival and the large number of resulting transformants, CRISPR-driven editing could enable the generation of edits normally challenging if not off-limits in bacteria, such as the generation of large libraries or simultaneous multiplexed edits. In contrast, efficient DNA targeting leads to rapid cutting of all genomic copies, forcing cells to undergo efficient homologous recombination immediately or eliminating cells that did not undergo recombination. Under these conditions, efficient DNA targeting would also select for escape mutants in the population that are resistant to CRISPR targeting (e.g., mutated Cas9 or gRNA plasmid, mutated target site)[53-55], accounting for the sometimes large fraction of unedited cells with a standard gRNA.

Achieving CRISPR-based editing in bacteria draws helpful parallels to editing in mammalian cells. There, DNA cuts engage different DNA repair pathways, allowing CRISPR to drive editing. In mammalian cells though, repair is dominated by end-joining pathways that lead to seemingly random edits. When relying on these pathways (e.g., for gene disruption), editing has proven to be highly efficient and can be readily multiplexed to disrupt numerous genes in one pass[56]. Achieving precise edits through homologous recombination in mammalian cells however requires either extensive screening or a series of approaches to enhance this repair pathway or inhibit other repair pathways[57-60]. As bacteria broadly lack the necessary machinery for end joining[61], homologous recombination instead represents the dominant repair pathway. For this reason, attenuating DNA targeting to enhance editing is likely unique to bacteria, despite the parallels to CRISPR-driven editing in human cells.

One challenge with attenuating DNA targeting in bacteria is finding the "sweet spot" that sufficiently drives RecA-mediated homologous recombination. If targeting is too strong, then a large population of cells will be killed off; if targeting is too weak, then few of the cells will undergo editing. Our work and prior work suggest that the extent of attenuation needed to hit this sweet spot likely varies between gRNA:target site pairs as well as organisms[15]. Fortunately, a large set of options are available to attenuate DNA targeting as illustrated in this work, with atgRNAs posing the most flexible and fine-tuned means. In particular, we found that adding hairpins to Cas9's sgRNA or modifying the repeat for Cas12a's gRNA each represents the simplest gRNA modifications without needing to consider the target site or sequence. However, introducing G:U wobbles or other mismatches between the guide and target site proved to be a relatively dependable means of achieving CRISPR-driven editing with both Cas9 and Cas12a. The caveat is that new mutations would need to be created for each new target site. It is also possible that an unmodified sgRNA in itself exhibits poor targeting activity and thus would promote CRISPR-driven editing. To find the most appropriate option, we recommend first testing representative atgRNAs (e.g., target mismatches, repeat mutations, gRNA format) in the WT and *recA*-deletion strain to identify modifications that greatly boost colony counts in the presence of *recA*; if a *recA* deletion cannot be obtained in the strain-of-interest, a dominant-negative RecA or homologous recombination inhibitor (e.g., GamS) can be co-expressed to inhibit the endogenous repair pathways[14,62]. Furthermore, we reason that, upon transformation with the selected mode of attenuated targeting and the RT, further culturing of colonies should continuously boost editing efficiencies by giving cells more opportunities to repair DSBs using the RT. In the future, we envision high-throughput screens combined with machine learning to predict the best atgRNA (or set of

atgRNAs) for a given site and desired edit, paralleling work developing efficient guides for different applications[63-65].

Once an appropriate mode of attenuated targeting is identified, editing would follow a series of steps paralleling current use of traditional CRISPR-based editing techniques in bacteria. First, the designed atgRNA and repair template would be cloned into a plasmid construct–ideally with the CRISPR nuclease to generate an all-in-one plasmid. To ensure all constructs are removed from the edited strain for downstream use, the plasmid could be encoded with an origin-of-replication that is temperature-sensitive or can be easily cured. Non-replicating repair templates such as an oligonucleotide, linear DNA, or a non-replicating plasmid could also be used, although the editing efficiency with attenuated targeting will likely be reduced because the repaire template would not be maintained. A recombinase system, such as λ-*red*, can also be introduced to further improve or even achieve editing, as we observed for large deletions in *K. oxytoca*. Given the utility of attenuated targeting in *E. coli* and in *Klebsiella* species, we anticipate that our approach could be broadly applied across the bacterial world, facilitating future mechanistic studies and engineering efforts.

## Methods

### Plasmid and strain construction
Supplementary Data 1 contains all strains, plasmids, and oligonucleotides used in this work. The *E. coli* MG1655 Δ*recA* (*recA*) and MG1655 Δ*lacI-Z* (*lacI-Z*) strains were produced using Flp-FRT recombination to remove the FRT-flanked *kanR* cassette in MG1655 Δ*recA::kanR* and MG1655 Δ*lacI-Z::kanR* intermediate strains, respectively. All plasmids were constructed using standard cloning techniques, with all information in Supplementary Data 1.

### Antibiotics
For all experiments in *E. coli* and *K. pneumoniae*, ampicillin (amp) was used at a final concentration of 100 μg/mL, kanamycin (kan) was used at a final concentration of 50 μg/mL, and chloramphenicol (cm) was used at a final concentration of 34 μg/mL. For all experiments in *K. oxytoca*, apramycin was used at a final concentration of 60 μg/mL and spectinomycin was used at a final concentration of 300 μg/mL.

### Genome targeting transformation assay in *E. coli*
Cells expressing Cas9 (Figs. 1, 2, 4 and Supplementary Figs. 1, 3), or AsCas12a (Figs. 3, 5 and Supplementary Figs. 2, 3) were used to assess genome targeting in MG1655 and MG1655 Δ*recA*. The strains were struck to isolation on a Luria-Bertani (LB) plate with the appropriate antibiotics. Single colonies were inoculated into liquid LB with antibiotics for overnight culturing at 37 °C shaking at 220 rpm. Each colony represents a biological replicate. The following morning, cells were back-diluted 50-fold into liquid LB with antibiotics and grown to an ABS$_{600}$ of 0.6–0.8. Subsequently, the cells were made electrocompetent and transformed via electroporation with 50 ng of the appropriate gRNA plasmid (see Supplementary Data 1 for more details). Cells were recovered in 500 μL of Super Optimal broth with Catabolite repression (SOC) medium for one hour at 37 °C shaking at 220 rpm and then plated in serial dilutions of 5-μL spots or using full plate dilutions on LB agar with appropriate antibiotics. The cells were incubated for 16 h at 37 °C. The number of colonies were counted the following morning and the number of transformants was back-calculated. One to two technical replicates of the cell dilutions were plated for the genome targeting assay to determine the number of transformants. The limit of detection was determined based on the volume of cells plated per condition.

### Editing assay in *E. coli*
To evaluate the editing efficiencies in *E. coli*, the same procedure was followed as in the transformation assay with small modifications. Repair templates (RT) were introduced into the gRNA plasmids with

250-bp upstream and downstream homology arms flanking the edit unless otherwise stated (i.e., Fig. 3e where 500-bp and 750-bp upstream and downstream homology arms were also tested). The RTs were designed such that the PAM would be modified while introducing a restriction enzyme site that would either result in silent mutations or an in-frame 6 bp insertion (i.e., introducing 5′-CCTAGG-3′, the AvrII recognition site, directly downstream of the *lacZ1* target site) to preserve functional *lacZ* upon editing. Therefore, different restriction enzymes were used to either introduce silent mutations for the particular target site or an in-frame small insertion to retain functional *lacZ*. The cells were made competent at an $ABS_{600}$ between 0.6–0.8 for experiments presented in Figs. 2b, 3c–e, 5c and Supplementary Figs. 1b, d, g, 2a–c. The cells were made competent at an $ABS_{600}$ of 1.4–1.6 for experiments presented in Figs. 1d, 2c, e, f and Supplementary Figs. 1f, 2d. Three to four technical replicates of the cell dilutions were plated for editing to ensure enough transformants would grow on the plates for subsequent screening. After transformation and selection on appropriate antibiotic plates, ten colonies from cells transformed with targeting gRNA plasmids and three colonies from cells transformed with the non-targeting gRNA plasmids were picked at random and re-struck onto LB plates with the appropriate antibiotics. These re-struck cells were incubated at 37 °C for 16 h. Colony PCRs were performed on the re-struck colonies using primers that flank the genomic target and anneal to the genomic DNA outside of the homology arms (i.e., prKC007 and prDC549 for experiments using WT or high-fidelity Cas9 nucleases targeting *lacZ* in *E. coli*, and oEV-775 and 776 or oEV-917 and 918 for experiments using AsCas12a to respectively insert small or bigger edits in *E. coli*). For editing experiments introducing a restriction enzyme site, PCR amplicons were then verified on a gel and purified with a PCR cleanup kit (DNA clean & concentrator 5, Zymo, Cat. no. D4014). 250 ng of the cleaned PCR products were then subjected to digestion by the appropriate restriction enzyme to evaluate if precise editing occurred. The digested products were resolved on a 1.5% agarose gel (or 1% for the Cas12a experiments) in Tris-acetate EDTA (TAE) buffer run at 80 V for 40 min and subsequently stained in ethidium bromide. The presence of any discernible digestion bands resulted in the colony being considered "edited". No obvious digestion bands resulted in the colony being considered "non-edited". For Cas12a experiments introducing a 2-bp deletion or bigger edits, 5 μL of the PCR products were directly loaded on the 1% agarose gel following the same conditions as described above for Cas9.

AvrII (NEB, Cat. no. R0174L) was used to evaluate precise editing for the RT used in Fig. 1d. BsgI (NEB, Cat. no. R0559L) was used to evaluate precise editing for the RT used in Fig. 2b, c, e. AclI was used to evaluate precise editing for the RT used in the left panel of Fig. 3c. RE digests were set up according to the manufacturer's protocol. Sanger sequencing followed by alignment to the original sequence was performed to evaluate editing for the RT used in the right panel of Fig. 3c when deleting 2 nts. Colonies were considered edited when at least 700 bp aligned perfectly to the reference genome but with the two missing base-pairs Colony PCR was used to evaluate precise editing for the RTs used in Fig. 3e when deleting the *lacZ* gene or replacing *lacZ* with *gfp*. The colonies for the lacZ deletion were considered edited when the colony PCR would result in the intended ~3.1-kb deletion compared to the unedited NT control. The colonies for the *gfp* substitution were considered edited when the colony PCR would result in the intended ~2.4-kb deletion compared to the unedited NT control. A non-selective out-growth was used for the aTc inducible sgRNA editing experiment, Fig. 2b. The transformation assay was performed as described above (see 'Genome targeting transformation assay in *E. coli*') through the recovery step in SOC. Upon recovery, 20 μL of the recovering cells were back-diluted 100x into non-selective outgrowth medium (LB with chloramphenicol and the appropriate amount of aTc) and incubated at 37 °C shaking at 22 rpm for 16 h. The cells were

then serially diluted in 1x PBS and 5-μL aliquots of each dilution were plated on LB plates with chloramphenicol and kanamycin without further aTc induction.

## Editing assay in *K. oxytoca*

*Klebsiella oxytoca* MK01 overnight cultures were back-diluted 50-fold in LB medium and subsequently grown to $ABS_{600}$ of 0.6–0.8. Cells were then made electrocompetent and transformed with 50 ng of pCas9KP plasmid. 50 μL competent cells were recovered with 950 μL liquid LB for one hour at 30 °C shaking at 220 rpm. The culture was centrifuged at $8000 \times g$ for 3 min. After the supernatant was removed, cells were resuspended with 50 μL of LB and plated on LB plates supplemented with 60 μg/mL apramycin. Three single colonies representing biological replicates were inoculated in LB with apramycin and cultured at 30 °C for 16 h. 1 mL of overnight cultures were inoculated into 100 mL of fresh LB and grown until $ABS_{600}$ of 0.15–0.2, then induced with 10 mL of 20% (w/v) L-arabinose LB supplemented with apramycin during which cultures were kept at room temperature shaking at 50 rpm for 2 h ($ABS_{600}$ of 0.6–0.8) and subsequently made electrocompetent as described above in the same paragraph.

To evaluate editing efficiencies presented in Fig. 4b, L-arabinose induced competent cells were co-transformed via electroporation with 200 ng of sgRNA plasmids and ≈500 ng of linear repair template assembled with 750 bp of upstream and downstream homology arms adjacent to the gene-of-interest using SOE-PCR. Cells were recovered with 950 μL of LB for one hour at 30 °C shaking at 220 rpm and then plated in serial dilutions on LB supplemented with apramycin (60 μg/mL) and spectinomycin (300 μg/mL). The limit-of-detection was determined based on the volume of cells plated per condition. After overnight incubation at 30 °C, 10 and 3 colonies were randomly picked from targeting sgRNA and non-targeting sgRNA containing transformation plates respectively and streaked out to fresh LB plates with the appropriate antibiotics and incubated at 30 °C for 16 h. The following day, colony PCRs were performed on the streaked-out colonies using primers targeting the genomic region outside of the homology arms (Proxy_22 and Proxy_31). Successful gene deletion was verified on 1% TAE agarose gel run at 130 V for 15 min and subsequently editing efficiencies were quantified.

## Editing assay in *K. pneumoniae*

The transformation assay from *E. coli* was followed with small changes for improving transformation efficiency in *Klebsiella pneumoniae* ATCC 10031. Specifically, cells expressing the AsCas12a nuclease codon optimized for this bacterium have been back-diluted 50-fold in LB medium with chloramphenicol and 0.7 mM EDTA and made competent at $ABS_{600} \approx 0.4$. The plasmid encoding the gRNA has been transformed as previously stated. For the editing experiment, a recombineering template with homology arms of 250 bp was cloned into the AsCas12a gRNA backbone prior to transformation. Three technical replicates of the cell dilutions were plated for editing to ensure enough transformants would appear on the plates for subsequent screening. The limit of detection was determined based on the volume of cells plated per condition. Ten and three colonies for the targeting and non-targeting samples respectively were picked for each biological replicate to be struck out on agar plates with chloramphenicol and kanamycin for subsequent screening by colony PCR and Sanger sequencing. Some of these colonies were struck out on ampicillin plates in parallel to check for resistance or sensitivity to ampicillin respectively for unedited and edited colonies, Fig. 5d. One non-edited colony transformed with the NT gRNA plasmid and one edited colony obtained by using the atgRNA15 plasmid were made competent and transformed with pAmpR (i.e. CBS-3946) or with water and plated either on ampicillin or LB-only agar plates. Colony PCR was performed for three edited colonies transformed with pAmpR and three transformed with water. 10 μL of the purified PCR products were

run on a 2% agarose TAE gel at 80 V for 40 min and stained for 15 min in EtBr prior to visualization.

## Northern blotting analysis

MG1655 Δ*lacI-Z* cells were transformed with pCas9 and the appropriate gRNA plasmid (Supplementary Data 1; pDC786, pDC841, pDC876, pDC869, pDC829, pDC860, pDC889, pDC862, and pDC886) and selected for on appropriate antibiotics. Individual colonies were inoculated into LB with appropriate antibiotics and incubated at 37 °C shaking at 220 rpm overnight. Cultures were then back-diluted into 15 mL of fresh LB with antibiotics and grown to an ABS$_{600}$ of 0.6–0.8. 5 mL of this culture was mixed with 1 mL of STOP mix, 95% of 100% ethanol and 5% hot phenol (Carl Roth Cat No. A980.1). The cultures were then snap-frozen in liquid nitrogen and stored at −80 °C until analysis. The frozen cultures were then thawed on ice for 1 h. The cultures were spun down at 4700 rpm at 4 °C for 20 min. The supernatant was removed and then subsequently spun down at 4700 rpm at 4 °C for 1 min. The supernatant was removed and the pellet was mixed with 600 μL of lysozyme-solution (0.5 mg/mL in TE buffer, pH 8.0). 60 μL of 10% SDS was added and incubated at 64 °C for 1 min. 66 μL of 1 M sodium acetate (pH 5.2) was mixed into the solution and then 750 μL of phenol (Carl Roth, Cat. no. A980.1) was added. The samples were incubated at 64 °C and vortexed briefly every 30 s for 6 min. The samples were then incubated on ice for 3–5 min and spun down at 13,000 rpm at 4 °C for 15 min. The upper phase was then transferred into the 2-mL PLG tube (QuantaBio, Cat. no. 2302830). 750 μL of chloroform (Carl Roth, Cat. no. 3313.1) was then added and incubated at room temperature for 3 min. The samples were then spun down at 13,000 rpm at 12 °C for 15 min. The upper phase was then transferred into a sterile 2 mL Eppendorf tube where 1.4 mL of 30:1 ethanol: sodium acetate pH 6.5 was added. The RNA was precipitated at −20 °C for 2 h. The samples were then spun down at 13,000 rpm at 4 °C for 30 minutes. The supernatant was removed and 500 μL of 70% ethanol was added to the pellet. The samples were spun down at 13,000 rpm at 4 °C for 10 min. The supernatant was removed and pellets were allowed to air dry at room temperature with the lid open. The pellets were dissolved in 75–100 μL nuclease-free water and allowed to incubate at 70 °C shaking for 5 min. 10 μg of each RNA sample were mixed with 2x GLII loading buffer (NEB, Cat. no. B0363A) loaded on an 8% polyacrylamide (Rotiphorese® Gel 40 (19:1) Acrylamide and Bisacrylamide solution Carl Roth, Cat. no.#3030.1) 7 M Urea gel. The samples were run on the gel at 300 V for 135 minutes using a gel transfer system (Doppel-Gelsystem Twin L, PerfectBlue). The samples were transferred to a Hybond-XL membrane (GE healthcare, RPN203S) at 50 V for 1 h at 4 °C and then cross-linked using 0.12 J (UV-lamp T8C; 254 nm, 8 W). The membrane was hybridized overnight in 15 mL of Roti-Hybrid Quick buffer (Carl Roth, Cat. no. A981.1) at 42 °C with 2.5–10 pmol/μL of γ −32P end-labeled oligodeoxyribonucleotides (Supplementary Data 1), then wrapped into a clear foil and exposed in the cassette with the phosphor screen for 3 days and visualized on a Phosphorimager (Typhoon FLA 7000, GE Healthcare). prDC1416 is the *lacZ1*-specific probe used in Fig. 1c, prDC1417 is the *lacZ2*-specific probe used in Supplementary Fig. 2c, and prDC1418 is the *lacZ3*-specific probe used in Supplementary Fig. 2c.

## Quantifying gRNA abundance

The raw northern blot images were used to quantify the relative abundance of fragments assumed to function as gRNAs. To calculate the abundance, ImageJ (https://imagej.nih.gov/ij/index.html) was used to measure the "mean gray value" for defined bands on the blot in each sample. We selected visible bands that were below 110 nts, as this was expected to represent either mature sgRNAs or crRNAs. Using ImageJ, a rectangle was drawn around a visible band and the "mean gray value" was measured. Then, the rectangle was moved horizontally to measure

the mean gray value for the other samples at the same location. In total, six rectangles were used for a given sample for *lacZ1*, three rectangles for *lacZ2*, and six rectangles for *lacZ3*, covering all discernable bands appearing below 110 nts. This was done for each sample on the blot until each sample had a value measured for that particular RNA fragment size (e.g. ~110 nts). This process was repeated for each band size observed below 110 nts for the northern blot with a *lacZ1*, *lacZ2*, and *lacZ3* probe. We used the same method of quantifying the mean gray value for the 5 S bands, which were used to normalize the quantified gRNAs from each sample. Following normalization to the 5 S, we summed the normalized gRNA abundance for each band size below 110 nts, and that is the value reported for each sample in the lower panel of Supplementary Fig. 1b.

## Statistics and reproducibility

No statistical method was used to predetermine sample size. No data were excluded from the analyses. The experiments were not randomized. The Investigators were not blinded to allocation during experiments and outcome assessment.

## Reporting summary

Further information on research design is available in the Nature Portfolio Reporting Summary linked to this article.

## Data availability

The original gel images, including all replicates, generated in this study have been deposited in Mendeley data under accession code[66]: https://doi.org/10.17632/f8ksz2nhd2.1. Source data are provided as a Source Data file, which also includes all original gel images as well as additional replicates. Source data are provided with this paper.

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

## Acknowledgements

The authors thank Atul K. Singh for generating the *E. coli ΔrecA* strain used in this work. The pCas9 and pCRISPR plasmids were gifts from Luciano Marraffini (Addgene #: 42876 and 42875 respectively). The pCP20 plasmid was a gift from Jörg Vogel. The plasmid carrying the *ampR* gene used in the *K. pneumoniae* experiment was kindly cloned by Katharina G. Wandera and was sent to Addgene (# 184844) for a separate publication. This work was funded by the Joint Programming Initiative on Antimicrobial Resistance (01KI1824 to C.L.B. and T.S.) and the DF-AMR2 funding initiative (project DECOLONIZE, FKZ: 01KI2131).

## Author contributions

D.C., E.V., and C.L.B. devised the study. D.C., E.V. J.Y., K.C., E.A., A.R., and T.A. performed experiments. D.C. and C.L.B. wrote the manuscript with input from all other authors. E.V. generated the figures. T.S. and C.L.B acquired funding and supervised the project.

## Funding

## Competing interests

C.L.B. is a co-founder and member of the scientific advisory board of Locus Biosciences and a member of the scientific advisory board of Benson Hill. The other authors declare no competing interests.
