## [Peer Review File · Nature Communications]

Reviewers' Comments:

Reviewer #1:

Remarks to the Author:

Reviewer comments NCOMMS-22-36117

Collias et al present methods to improve CRISPR-based editing in bacteria by reducing the targeting efficiency of the CRISPR system used. Current methods using CRISPR technology and/or recombineering in bacteria have low transformation efficiencies making it difficult to screen for desired edits. The authors tested a variety of modes to reduce CRISPR targeting activity such as guide (g) RNA format, gRNA expression promoter, Cas nuclease (types, WT and engineered variants), sub-optimal gRNA modifications, and homology arm length variations in the repair template. The authors conclude that using attenuated gRNAs or atgRNAs is the most tunable approach to scale Cas targeting activity. This was also demonstrated in clinically relevant *Klebsiella* species with the deletion of *cydAB* operon and ampicillin resistance marker. While the use of atgRNAs is an easy means to modulate editing activity, multiple variations of atgRNAs may have to be tested to get the desired levels of transformation efficiency and editing outcome. atgRNAs offer the potential to improve editing in bacteria and expand the genetic toolkit in microbiology. I believe that the manuscript is suitable for publication if the following comments are addressed with revision of the paper.

Major comments:

1. For all colony PCRs performed to check editing efficiency in this study, were the primers designed to anneal outside the homology arms (HA)? Primers designed to bind within HAs may lead to false positives from amplification of the repair template plasmid in the agar. Please elaborate in the methods how the PCR assays are designed.

2. In all instances where a repair template was used, (for example, line 102, 193), please specify the repair template features such as insert size, homology arm size, etc. Please include this in the methods as done in some cases (lines 368, 383).

a. Line 341: Why were different RE used for the different RT to measure editing efficiency? This may become clear if the template designs are elaborated.

3. There are some additional testing opportunities that may fill gaps in this study. If the authors have a suitable explanation for these, experiments may not be necessary.

a. Was sgRNA with Pn tested? Do the authors expect the sgRNA expression to be different and potentially result in better transformation efficiency?

b. Were atgRNAs tested with high fidelity (HF) Cas9 variants? Would this drive towards higher transformation and editing efficiencies?

c. In experiment testing the different HF Cas9 variants, can the authors attest to the expression levels of the different variants? Can the difference in the transformation and editing efficiencies stem from varying Cas9 variant expression?

4. There are some inconsistencies in the choice of DNA targeting modulation method used throughout the study that also disrupts the flow from one section to another. Please clarify these conflicts in the results section:

a. In the experiment to test inducible promoter (fig 2B), why was sgRNA tested and not crRNA? sgRNA was also chosen for other experiments in this study. From fig 1B and line 75, crRNA constructs had much better transformation efficiency.

b. In fig 4B, why was EHP2 sgRNA used and not MM4 sgRNA. From fig 2E, MM4 had higher transformation and editing efficiencies. What promoter was used here for gRNA expression?

c. Why was kpMM4 (fig S3E) left out of the final 3 guides (Fig 5B) tested?

5. The authors are missing some citations that will better support their study:

a. Line 60: please cite (PMID: 22745249 PMID: PMC6286148 DOI: 10.1126/science.1225829) when introducing sgRNA.

b. Line 142: It is worth citing (PMID: 32187529 PMID: PMC7370240 doi: 10.1016/j.molcel.2020.02.023) and (PMID: 33744974 PMID: PMC8053117 DOI:

10.1093/nar/gkab163) when discussing the reduced activity of HF Cas9 variants.

c. The authors tested atgRNAs with more than 2 mismatches (MM3, MM4) and only cite Jones et al 2021 Nat Biotechnol (ref 22) where testing was limited to targets with up to 2 MM. It would be more appropriate to also cite these research papers: (PMID: 33744974 PMID: PMC8053117 DOI: 10.1093/nar/gkab163); (PMID: 32161115 PMID: PMC7186167 DOI:

10.1074/jbc.RA120.012933); (PMID: 34162850 PMID: PMC8222333 DOI: 10.1038/s41467-021-24017-8); (PMID: 32315032 PMID: PMC7229833 DOI: 10.1093/nar/gkaa231).

d. A recent study tested mismatched sgRNAs to introduce point mutation in bacteria, although in combination with lambda-red recombineering (PMID: 32327447 PMID: PMC7263196 doi: 10.1101/gr.257493.119). This should be discussed along with Wang et al 2018 Appl Env Microbiol (ref 40) in line 208.

6. While the Excel file listing all the plasmids and guides, especially with the benchling links is helpful, it would be more beneficial for readers to see a supplementary table with different atgRNAs tested. The table could have the name (example, MM4, EHP1, etc), type of modification (UE, NC, etc), target gene, promoter, the sequence with the modification (mismatch, PAM, extensions, etc) highlighted, and the associated data figure. This would also resolve the following issues:

a. Line 158: typo on 3 or 4 MM: "which three target mismatches...sgRNA guide (MM4)".

b. Fig S2C – what is MM1 in panel on the far right? Should it be SM1?

c. In fig 3B, is it atgRNA6 mutR5 or mutR6 since mutR5 does slightly better than mutR6 in fig S2B.

d. Please maintain atgRNA nomenclature consistency throughout text. For example, line 247: atgRNA3 in text vs atgRNA_kp3 in figure.

7. In fig 3B, C, a 5 MM guide (atgRNA1 – MM5) was used to reduce Cas12a cleavage activity. It was recently shown that Cas12a has off-target nicking activity particularly against target with multiple mismatches in (PMID: 30833733 PMID: PMC6512873 DOI: 10.1038/s41564-019-0382-0) and (PMID: 32161115 PMID: PMC7186167 DOI: 10.1074/jbc.RA120.012933). When using guides with multiple mismatches, Cas12a is more likely to nick rather than create DSBs. Can the authors comment on how nicks are repaired in bacteria and if the RecA dependent pathway is the likely mode? In the case of Cas12a atgRNAs that have multiple mismatches, would nicking rather than DSB formation be the mode of reduced DNA targeting activity?

8. For the different atgRNA modifications tested in fig 2 D, E, there is about a 10- to 100-fold difference in colony formation. Is this difference due to the ABS600 at which the cells were made competent (lines 328 – 330)? What were the other major changes between the experimental set-up in panels D and E? Is this variability expected between experiments?

a. Line 655: It may be worth discussing this in the results

9. Line 337: How was the editing efficiency quantified? Was only the presence of "edited" band counted for quantification (for example, fig S1E – ps sgRNA lane 2 vs 3) or was the intensity of the WT and edited bands (for example, fig S2E, atgRNA1 – lanes 4 vs 5) considered? What numbers are plotted in the graphs? Please elaborate in methods.

a. For all agarose gels, please indicate the WT and edited band sizes i.e. WT PCR amplicon, RE cut products size or edited PCR amplicon sizes

10. Line 226: what other variants of atgRNA would have been tested if EHP2 did not work? From the testing all different atgRNAs, can the authors compile top variants that may work across Cas nucleases and targets as a starting point for researchers who may want to implement this method in their model bacterial species?

Minor suggestions:

1. It would be helpful for the readers if the authors called out specific examples and data points in the results throughout the text (for example, for lines 67 to 72) as done in line 158.

2. There are several instances where the target gene information is missing in the results sections and/or figures. To make it easier for the readers, please consider including the site information within the figures (as done in fig S1B)

- a. Fig 1D, add site label in figure too?
 - b. What target site was tested in fig 2B and 2C?
 - c. Some figures panels have the gene in the figure while others don't – fig S1F NC PAM targets, fig S2C.
3. Lines 139, 240: For consistency, the authors could indicate "constitutive synthetic sgRNA" as Ps sgRNA in parenthesis.
 4. Line 138: Do the authors mean HypaCas9 when they refer to SpyCas9_HeF? If the authors, choose to call this variant HeF then please cite the appropriate publication after each HF Cas9 in the text for clarity.
 5. Please define abbreviations:
 - a. Line 148: SM and MM.
 - b. Line 73, 81: RSR and SR.
 6. Line 96: Why do the crRNA sizes vary. Is this due to improper processing?
 7. Line 140, 142: Please indicate Cas9 here was SpyCas9 to be consistent with fig 2C labels.
 8. Line 315: please include concentrations of the antibiotics used in all instances, either in the methods or supplementary table.
 9. Line 317: what was the back-dilution ratio? Add this detail throughout the methods
 10. Fig 1C, consider including labels (RSR, SR, sg) to the figures on top of the blot
 11. Fig 1B, lower panel – is there a reason kefC is missing from Ps RSR?
 12. Fig1B, it may be worth splitting this into two panels 1B for top graphs and 1C for bottom graphs and linking to paragraph in line 79 to 88.
 13. Fig S1A, add figure legend to indicate these data are WT only.
 14. Fig 2B, consider adding NT, T labels. And in other figure panels where they are missing.
 15. Fig 5B, are the figures for MM2 and MM3 swapped?
 16. Line 344: was sanger sequencing performed only for the 2 nt deletion experiments? Is there a possibility of improper HR that can introduce SNPs in the HA?
 - a. line 259: "precise genome editing": unless the edited samples were also sequenced to confirm then nature of RT insertion, it may be inaccurate to use the term "precise". Colony PCRs performed in this study only show the presence of the edit at the intended target site.
 17. Line 381: "back-diluted in LB medium with cm and". what is "cm"?
 18. Line 338: Please include agarose gel conditions throughout methods as done in lines 375, 393

Reviewer #2:

Remarks to the Author:

Summary

In this manuscript, the authors describe an approach to improve CRISPR-driven genome editing in bacteria by systematically attenuating DNA targeting activity. Previous methods require DNA recombinases and Cas nucleases in tandem and are often characterized by low transformation and editing efficiencies. Here, the authors found that perturbing CRISPR activity, typically by modifying gRNA structure or expression levels, can increase colony counts and recombination efficiency in

CRISPR-driven editing. These improvements removed the need for exogenous recombinases in *E. coli* and enabled editing in several non-model bacterial strains. The new approaches described in this manuscript will be of broad interest to the bacterial gene editing and synthetic biology communities, and are well-suited for publication after the authors consider the comments below.

Major Comments

1. On line 259, the authors describe a general model to explain their results: "This seemingly paradoxical concept--making DNA targeting weaker can improve editing--appears to emerge from the cells having time to repair cut genomic DNA with other copies of the chromosome or a provided RT." Can the authors elaborate and discuss this point further? Once a cut happens, presumably the cell needs to repair it before replication, otherwise it dies. How does attenuating DNA targeting affect the time window between cleavage and replication? Or is the idea that rapidly dividing cells have multiple copies of the genome and efficient cutting will cleave all copies so the cells die if repair does not occur, but attenuated cutting means some cells will survive and the population will have multiple attempts through multiple cell cycles to introduce the edit? Or could the observed effects arise if the attenuated CRISPR complexes more readily dissociate from DNA after cleavage, allowing the repair machinery easier access?

2. Can the authors address how this editing system would work in practice for engineering edited strains? Specifically, would plasmid curing be used to remove the CRISPR system components and/or would non-replicating plasmids be effective? Additional discussion would be sufficient, no new experiments are needed.

Minor comments

1. In the introduction line 30 it took a few reads to parse and understand this sentence: "Despite the paradigm of utilizing chromosomal cleavage to counterselect against unedited cells, prior work reported an intriguing exception: repair of chromosomal cleavage by Cas9 in *Escherichia coli* through endogenous homologous recombination." In contrast, the same point on line 54 was very clear and immediately understandable: "We were initially intrigued why targeting some locations in the *E. coli* genome with the *Streptococcus pyogenes* Cas9 led to RecA-dependent homologous recombination rather than cell death." The authors might consider rephrasing the statement on line 30.

2. The authors used Cas9 for *K. oxytoca* and Cas12a in *K. pneumonia*. Is there a reason different Cas systems were used and are there general considerations that guided this decision?

3. The labels on Figure 1C are inconsistent with the labels in Fig 1B&D, making it difficult to compare between panels. It would also be helpful if the authors could identify the specific bands corresponding to each expression construct. Can any of the higher molecular weight bands be identified?

4. On line 93, the authors write: "Northern blotting analysis on the complete set of gRNAs targeting *lacZ1* revealed an inverse correlation between the abundance of the final gRNA (Ps sgRNA > Ps SR > Pn SRS \approx Pn SR > Ps RSR) with colony counts in the presence of *recA*." Can the authors provide a quantitative analysis based on band intensities in the Northern blot?

5. On line 187, the authors write: "High colony counts and editing was lost in the absence of *recA* (Fig. S2F), confirming the importance of RecA-mediated homologous recombination with atgRNAs." Fig S2F is an important control that clearly supports the model that RecA-mediated repair is responsible for recombination. The authors could consider moving this point to a main text figure panel.

6. Several figure panels show a limit of detection for colony count assays, and this limit of detection appears to vary between figures. Can the authors clarify in the methods section how the limit of detection was calculated?

7. On line 215, the authors note that lambda red recombination was still necessary for editing in *K.*

oxytoca. It appears that lambda red was not necessary for editing in *K. pneumoniae*. Can the authors provide any guidance or predictions on when lambda red will be necessary? Would users need to experimentally test each new strain, or are there differences in endogenous recombinases that might be predictive?

8. On line 220, the authors write: "For the sp47 sgRNA, the atgRNA boosted colony counts to be consistently above the limit of detection and yielded an editing frequency of ~83% (Fig. 4B)." In the same figure, several unmodified sgRNAs appear to show relatively high editing efficiencies and colony counts, and the unmodified sp55 sgRNA appears to give the best combination of colony counts and recombination efficiency. Can the authors comment on this observation? Do the authors expect that if enough target sites are screened, suitable sgRNAs might be identified with high editing efficiency for other editing targets? Could the sp55 sgRNA be a potentially poor target site or misfolded sgRNA that is already partially attenuated?

Reviewer #3:

Remarks to the Author:

In this work, the authors demonstrated that systematically attenuating DNA targeting activity can achieve CRISPR-driven editing in bacteria, greatly boosting colony counts and even increasing the frequency of precise genome editing. However, the concept of attenuating Cas protein expression of gRNA expression/format is not new. The authors do not provide novel concept or design, nor do they demonstrate very solid data to prove its suitability to publish in Nature Communications. I would recommend rejection of this paper.

Other major critiques:

1. The manuscript is not well written in terms of clarity. The authors claim the use of attenuated gRNA. However, it is unclear what this is after reading the abstract and introduction.
2. Fig. 1, the effects of Rec A on recombination in *E. coli* is well known and has been studied in CRISPR previously.
3. The authors claim that this method obviates the need of recombinase. However, the use of recombinase or lambda red provides excellent genome editing efficiency and low off-target effects. The CRISPR-lambda red system also enable multiplexing and rational metabolic engineering of *E. coli*. The authors only provide some conceptual data without demonstrating its advantage over other approaches. It's unlikely that this approach will be used broadly.

Reviewer comments NCOMMS-22-36117

We thank all three reviewers for their consideration, input, and helpful comments. We address each comment below. Accompanying changes in the main text are in red.

In addition to new experiments and text changes, we have updated two datasets in the Supplementary Information (Figs. S3A and S3C). We were unable to find the original gel images used to quantify those data, so we repeated the experiments. The repeated experiments yielded the same trends as the original data and resulted in similar values.

Reviewer #1 (Remarks to the Author):

Collias et al present methods to improve CRISPR-based editing in bacteria by reducing the targeting efficiency of the CRISPR system used. Current methods using CRISPR technology and/or recombineering in bacteria have low transformation efficiencies making it difficult to screen for desired edits. The authors tested a variety of modes to reduce CRISPR targeting activity such as guide (g) RNA format, gRNA expression promoter, Cas nuclease (types, WT and engineered variants), sub-optimal gRNA modifications, and homology arm length variations in the repair template. The authors conclude that using attenuated gRNAs or atgRNAs is the most tunable approach to scale Cas targeting activity. This was also demonstrated in clinically relevant *Klebsiella* species with the deletion of *cydAB* operon and ampicillin resistance marker. While the use of atgRNAs is an easy means to modulate editing activity, multiple variations of atgRNAs may have to be tested to get the desired levels of transformation efficiency and editing outcome. atgRNAs offer the potential to improve editing in bacteria and expand the genetic toolkit in microbiology.

I believe that the manuscript is suitable for publication if the following comments are addressed with revision of the paper.

We thank the reviewer for their on-point summary of our manuscript and their extensive guidance on how to improve the work. We incorporated every suggestion, in some cases performing follow-on experiments to strengthen our claims.

Major comments:

1. For all colony PCRs performed to check editing efficiency in this study, were the primers designed to anneal outside the homology arms (HA)? Primers designed to bind within HAs may lead to false positives from amplification of the repair template plasmid in the agar. Please elaborate in the methods how the PCR assays are designed.

All primers that were used to evaluate editing efficiency were designed to anneal outside of the homology arms such that only the genomic DNA would be amplified. To clarify this point, we have made the following text changes in the Methods section (with changes shown in red):

“Colony PCRs were performed on the re-struck colonies using primers that flank the genomic target and anneal to the genomic DNA outside of the homology arms (i.e., prKC007 and prDC549 for experiments using WT or high-fidelity Cas9 nucleases targeting *lacZ* in *E. coli*, and oEV-775 and 776 or oEV-917 and 918 for experiments using AsCas12a to respectively insert small or bigger edits in *E. coli*.”

2. In all instances where a repair template was used, (for example, line 102, 193), please specify the repair template features such as insert size, homology arm size, etc. Please include this in the methods as done in some cases (lines 368, 383).

a. Line 341: Why were different RE used for the different RT to measure editing efficiency? This may become clear if the template designs are elaborated.

The reviewer raises a good question that we can more thoroughly describe. We sought to introduce silent mutations to insert restriction enzyme recognition sites at the particular target site (e.g. *AvrII* for *lacZ1* and *BsgI* for *lacZ3* with Cas9). In almost all instances, we used 250-bp upstream and downstream homology arms introduced on the gRNA plasmid. To better address these points in the manuscript, we have updated the methods section with the following:

“Repair templates (RT) were introduced into the gRNA plasmids with 250-bp upstream and downstream homology arms flanking the edit unless otherwise stated (i.e., Fig. 3E where 500-bp and 750-bp upstream and downstream homology arms were also tested). The RTs were designed such that the PAM would be modified while introducing a restriction enzyme site that would either result in silent mutations or an in-frame 6-bp insertion (i.e., introducing 5'-CCTAGG-3', the *AvrII* recognition site, directly downstream of the *lacZ1* target site) to preserve functional *lacZ* upon editing. Therefore, different restriction enzymes were used to either introduce silent mutations for the particular target site or an in-frame small insertion to retain functional *lacZ*.”

3. There are some additional testing opportunities that may fill gaps in this study. If the authors have a suitable explanation for these, experiments may not be necessary.

a. Was sgRNA with Pn tested? Do the authors expect the sgRNA expression to be different and potentially result in better transformation efficiency?

The sgRNAs were not tested with Pn. The reasoning was more technical: there is conflicting data on the location of the transcriptional start site for this promoter in *E. coli*, where wrong placement could impact the activity of the sgRNA and potentially bias our conclusions. That said, we agreed that it was worth exploring the impact of sgRNA expression levels on colony counts. We therefore varied sgRNA expression by replacing the original strong J23119 promoter with one of two different constitutive promoters (the medium J23116 promoter and the weak J23109 promoter), where the expression strength of each promoter was measured in our prior work (PMID =

35690065). Repeating the transformation assay, we found that increasing sgRNA expression resulted in few colony counts, in line with gRNA expression strength contributing to cell death.

The new data were integrated as Fig. S1D shown below.

We also added the following text on p. 6:

“To explicitly evaluate the impact of expression strength, we replaced the strong constitutive promoter in front of the *lacZ1*- and *loID*-targeting sgRNAs with two weaker versions: a medium promoter (J23116) and a weak promoter (J23109)²⁰. The weak promoter yielded similarly high colony counts ($\sim 10^5$ - 10^6 transformants) for WT and $\Delta recA$ cells for both sites indicative of cell survival, while the medium promoter yielded low colony counts (0 - $\sim 10^2$ transformants) in WT and $\Delta recA$ strains similar to the strong promoter and indicative of cell death (Fig. S1D).”

b. Were atgRNAs tested with high fidelity (HF) Cas9 variants? Would this drive towards higher transformation and editing efficiencies?

Initially we did not test atgRNAs with any of the Hi-Fi Cas9's. However, given the reviewers' good suggestion, we tested the eSpCas9(1.1) variant with some of the best performing atgRNAs identified in Fig. 2E (MM4, EHP1, IHP3, and UE7). We found that MM4 led to 100% editing with eSpCas9(1.1) and no drop in the number of transformants compared to the NT control. These results were added as Fig. S1F as shown below.

We further have added the following text on p. 9:

“Separately, atgRNA1 and atgRNA6 guiding the high fidelity eSpCas9(1.1) increased colony counts by $\sim 10^3$ -fold and yielded editing efficiencies of 100% and 50%, respectively (**Fig. S1F**). The colonies screened for atgRNA4 and atgRNA6 resulted in only faint digestion patterns (Fig. S1H), indicating that colonies exhibit a mixed genotype and may still be undergoing editing. In many of these cases, introducing the RT into the gRNA plasmid affected the number of transformants, although the ratio of colonies between non-targeting and targeting conditions were similar with or without the RT.”

c. In experiment testing the different HF Cas9 variants, can the authors attest to the expression levels of the different variants? Can the difference in the transformation and editing efficiencies stem from varying Cas9 variant expression?

We did not explicitly test the expression levels of the three high-fidelity Cas9 variants to compare it with that of the WT version. However, the constructs we used were nearly identical. Specifically, the high-fidelity mutations were cloned into *cas9* in the WT backbone plasmid such that the promoter and 5' UTR are the exact same. Therefore we expect that the expression levels should be very similar between these constructs but cannot say that with certainty. We agree with the review that lower expression could lead to attenuated activity.

4. There are some inconsistencies in the choice of DNA targeting modulation method used throughout the study that also disrupts the flow from one section to another. Please clarify these conflicts in the results section:

a. In the experiment to test inducible promoter (fig 2B), why was sgRNA tested and not crRNA? sgRNA was also chosen for other experiments in this study. From fig 1B and line 75, crRNA constructs had much better transformation efficiency.

We selected an sgRNA over the crRNA or tracrRNA, as encoding either the crRNA or tracrRNA would cause the other RNA to become a limiting component, restricting the tunability of targeting activity and possibly confounding interpretation of the results. To better explain this, we have added the following text:

“We further encoded an sgRNA, as encoding either a crRNA or tracrRNA could cause the other RNA to become limiting.”

b. In fig 4B, why was EHP2 sgRNA used and not MM4 sgRNA. From fig 2E, MM4 had higher transformation and editing efficiencies. What promoter was used here for gRNA expression?

We did not use MM4 in Fig. 4B because the associated guide mutations that generate G:U wobbles between the target strand and guide would not necessarily translate to the new target site. We instead sought to test an atgRNA method that was not target-site specific (like EHP, IHP, and UEs). Therefore, we added EHP2 to the sgRNAs without having prior knowledge whether this would lead to the best outcome. To clarify this in the text, we have added the following text on p. 12:

“When considering which sgRNA modifications to introduce, we opted to introduce the hairpin EHP2 to the PAM-distal end of each sgRNA because it can be introduced independently of the target sequence (**Figs. 4B and S3D**).”

c. Why was kpMM4 (fig S3E) left out of the final 3 guides (Fig 5B) tested?

kpMM4 was left out in the final test because it yielded a similarly high number of transformants between the WT and $\Delta recA$ strain using kpMM4 in Fig. S3E. Therefore we did not expect that it should lead to genome editing.

5. The authors are missing some citations that will better support their study:

a. Line 60: please cite (PMID: 22745249 PMCID: PMC6286148 DOI: 10.1126/science.1225829) when introducing sgRNA.

b. Line 142: It is worth citing (PMID: 32187529 PMCID: PMC7370240 doi: 10.1016/j.molcel.2020.02.023) and (PMID: 33744974 PMCID: PMC8053117 DOI: 10.1093/nar/gkab163) when discussing the reduced activity of HF Cas9 variants.

c. The authors tested atgRNAs with more than 2 mismatches (MM3, MM4) and only cite Jones et al 2021 Nat Biotechnol (ref 22) where testing was limited to targets with up to 2 MM. It would be more appropriate to also cite these research papers: (PMID: 33744974 PMCID: PMC8053117

DOI: 10.1093/nar/gkab163); (PMID: 32161115 PMID: PMC7186167 DOI: 10.1074/jbc.RA120.012933); (PMID: 34162850 PMID: PMC8222333 DOI: 10.1038/s41467-021-24017-8); (PMID: 32315032 PMID: PMC7229833 DOI: 10.1093/nar/gkaa231).

d. A recent study tested mismatched sgRNAs to introduce point mutation in bacteria, although in combination with lambda-red recombineering (PMID: 32327447 PMID: PMC7263196 doi: 10.1101/gr.257493.119). This should be discussed along with Wang et al 2018 Appl Env Microbiol (ref 40) in line 208.

We thank the reviewer for bringing these additional citations to our attention. All have been added as suggested.

We also note the mismatch naming SM and MM refer to a single mismatch and multiple mismatches, respectively. The numbering was meant to track the different tested mutations rather than the number of given mutations (e.g., MM3 is multiple mutation set #3). We clarify this with the following added text on p. 8:

“Fortunately, numerous modifications are known that can have a minor to massive impact on DNA targeting, including introducing single mismatches (SM) or multiple mismatches (MM) into the guide sequence at different locations, The proceeding number specifies the order in which the mutation was tested (e.g., MM3 for the third tested multiple-mismatch modification).”

6. While the Excel file listing all the plasmids and guides, especially with the benchling links is helpful, it would be more beneficial for readers to see a supplementary table with different atgRNAs tested. The table could have the name (example, MM4, EHP1, etc), type of modification (UE, NC, etc), target gene, promoter, the sequence with the modification (mismatch, PAM, extensions, etc) highlighted, and the associated data figure. This would also resolve the following issues:

a. Line 158: typo on 3 or 4 MM: “which three target mismatches...sgRNA guide (MM4)”.

We apologize for the confusion in our naming scheme. The number does not reflect the number of mismatches and rather is just sequential based on the order in which we designed them. MM4 means it's the fourth construct we tested with multiple mismatches. We have updated the text to clarify this:

“The proceeding number specifies the order in which the mutation was tested (e.g., MM3 for the third tested multiple-mismatch modification)”

b. Fig S2C – what is MM1 in panel on the far right? Should it be SM1?

The labeling is correct in the panel, but the order in which we plotted the data could be confusing. Therefore, we re-organized the data such that all of the single mismatched constructs (SM#)

appear grouped together and all of the multiple mismatched constructs (MM#) appear grouped together. The updated figure panel can be seen below:

c. In fig 3B, is it atgRNA6 mutR5 or mutR6 since mutR5 does slightly better than mutR6 in fig S2B.

Based on the results in Fig. S2B, we decided to test mutR5 and mutR6 in both WT and $\Delta recA$ strains in Fig 3A. Based on the difference between the colony counts in the WT compared to the $\Delta recA$ strain, we assigned mutR6 as atgRNA2 for testing with a RT.

d. Please maintain atgRNA nomenclature consistency throughout text. For example, line 247: atgRNA3 in text vs atgRNA_kp3 in figure.

We apologize for the confusion stemming from our original naming scheme. We have updated the atgRNA naming scheme to hopefully alleviate the confusion. Each atgRNA is now given a number based on the order in which they appear in the manuscript. For instance, atgRNA1 is the *lacZ3* sgRNA with the MM4 modification (see updated Fig. 2E). We have also added text into each figure to specify which mutations were used to generate the atgRNAs present in that figure.

7. In fig 3B, C, a 5 MM guide (atgRNA1 – MM5) was used to reduce Cas12a cleavage activity. It was recently shown that Cas12a has off-target nicking activity particularly against target with multiple mismatches in (PMID: 30833733 PMID: PMC6512873 DOI: 10.1038/s41564-019-0382-0) and (PMID: 32161115 PMID: PMC7186167 DOI: 10.1074/jbc.RA120.012933). When using guides with multiple mismatches, Cas12a is more likely to nick rather than create DSBs. Can the authors comment on how nicks are repaired in bacteria and if the RecA dependent pathway is the likely mode? In the case of Cas12a atgRNAs that have multiple mismatches, would nicking rather than DSB formation be the mode of reduced DNA targeting activity?

The reviewer raises an excellent point that Cas12a may be nicking rather than cleaving DNA targets with mismatches, offering a distinct means of driving homologous recombination. Cas9-driven DNA nicking is a poor driver of homologous recombination in *E. coli* in our experience,

although it has proven effective in other bacteria such as lactobacilli (e.g., PMID = 28864652). We therefore commented on the possibility of nicking driving homologous recombination in the discussion. Specifically, we added the following to p. 14:

“In addition, attenuated targeting may lead to nicking rather than cleavage of the DNA target^{28,51}, which can also be repaired through homologous recombination⁵².”

8. For the different atgRNA modifications tested in fig 2 D, E, there is about a 10- to 100-fold difference in colony formation. Is this difference due to the ABS600 at which the cells were made competent (lines 328 – 330)? What were the other major changes between the experimental set-up in panels D and E? Is this variability expected between experiments?

a. Line 655: It may be worth discussing this in the results

The constructs used in Fig. 2D only contain a gRNA, whereas the constructs used in Fig. 2E contain the gRNA and the repair template. In addition, all of the editing experiments with Cas9 in *E. coli* were performed by making the cells electrocompetent at an OD of 1.4-1.6 (Fig. 2E), whereas the transformation assays in *E. coli* were performed at an OD of 0.6-0.8. Typically, we see a lower transformation efficiency when repair templates are introduced, although the exact values depend on the size of the homology arms, the type of edit, and the orientation of the homology arms relative to the backbone of the construct. We agree that it is worth discussing this point and have added the following to p. 9:

“In many of these cases, introducing the RT into the gRNA plasmid affected the number of transformants, although the ratio of colonies between non-targeting and targeting conditions were similar with or without the RT. ”

9. Line 337: How was the editing efficiency quantified? Was only the presence of “edited” band counted for quantification (for example, fig S1E – ps sgRNA lane 2 vs 3) or was the intensity of the WT and edited bands (for example, fig S2E, atgRNA1 – lanes 4 vs 5) considered? What numbers are plotted in the graphs? Please elaborate in methods.

For editing experiments that introduced a restriction site, the presence of any discernible digestion band was considered “edited”. A colony was considered non-edited when there was no discernible digestion band. We have updated the methods to capture this with the following text update:

“The presence of any discernible digestion bands resulted in the colony being considered “edited”. No obvious digestion bands resulted in the colony being considered “non-edited”.”

To evaluate editing efficiencies, three non-targeting colonies were selected at random and 10 targeting colonies were selected per biological replicate. The edited versus non-edited colonies

for each biological replicate were plotted in the figures. This information can be found in the figure legends and methods section.

a. For all agarose gels, please indicate the WT and edited band sizes i.e. WT PCR amplicon, RE cut products size or edited PCR amplicon sizes

To clarify this point, we added more descriptions to Figs. S1E, S2E, S2E, and S3D to indicate whether the edited product was the result of restriction enzyme digest, a gene deletion, or a substitution. We also updated the figure legends and methods to provide more details about the size of amplicons or digested fragments.

10. Line 226: what other variants of atgRNA would have been tested if EHP2 did not work? From the testing all different atgRNAs, can the authors compile top variants that may work across Cas nucleases and targets as a starting point for researchers who may want to implement this method in their model bacterial species?

We could imagine other modifications that are not specific to a given target site (i.e., hairpins, extensions, etc.) to be suitable for random testing. We have seen that introducing mismatches can lead to a high number of transformants and editing efficiencies but this would need to be designed for each target site. We think this is a great point to better address in the discussion. We therefore added the following to the text on p. 15:

“In particular, we found that adding hairpins to Cas9’s sgRNA or modifying the repeat for Cas12a’s gRNA each represents the simplest gRNA modifications without needing to consider the target site or sequence. However, introducing G:U wobbles or other mismatches between the guide and target site proved to be a relatively dependable means of achieving CRISPR-driven editing with both Cas9 and Cas12a. The caveat is that new mutations would need to be created for each new target site. It is also possible that an unmodified sgRNA in itself exhibits poor targeting activity and thus would promote CRISPR-driven editing.”

Minor suggestions:

1. It would be helpful for the readers if the authors called out specific examples and data points in the results throughout the text (for example, for lines 67 to 72) as done in line 158.

The reviewer makes a great point about how to improve the overall accessibility of our writing. Following this suggestion, we went through the manuscript carefully and added values where appropriate. In the case of lines 67 - 72, these sentences are generally describing expectations for experiments with or without RecA before any experiments are performed. We realize the original wording suggested we were describing our experimental results, so we reworded the first sentence to read the following:

“Under this setup, cell counts of the WT and $\Delta recA$ strains are expected to depend on targeting activity....”

2. There are several instances where the target gene information is missing in the results sections and/or figures. To make it easier for the readers, please consider including the site information within the figures (as done in fig S1B)
 - a. Fig 1D, add site label in figure too?
 - b. What target site was tested in fig 2B and 2C?
 - c. Some figures panels have the gene in the figure while others don't – fig S1F NC PAM targets, fig S2C.

We thank the reviewer for pointing out this inconsistency. We have updated all of the figures to contain the target gene.

3. Lines 139, 240: For consistency, the authors could indicate “constitutive synthetic sgRNA” as Ps sgRNA in parathesis.

Following the reviewer's suggestion, we inserted more labels in the text to provide a better link to the figures. This change included inserting Ps sgRNA on p. 8 and the various gRNA variants on p. 12.

4. Line 138: Do the authors mean HypaCas9 when they refer to SpyCas9_HeF? If the authors, choose to call this variant HeF then please cite the appropriate publication after each HF Cas9 in the text for clarity.

We thank the reviewer for bringing this up, as this could be confusing for readers. We are referring to the HeF variant that was mentioned and tested in doi: 10.1038/nature24268 and in doi: 10.1186/s13059-017-1318-8. The variant contains the combined mutations from the eSpCas9(1.1) and HF1. To clarify this, we added the following to p. 8:

“We found that SpyCas9_HeF, which contains the combined mutations of both eSpCas9(1.1) and SpyCas9_HF122,24, exhibited much higher colony counts than the parental Cas9 (SpyCas9) and the other two variants ($\sim 10^3$ -fold)....”

5. Please define abbreviations:
 - a. Line 148: SM and MM.
 - b. Line 73, 81: RSR and SR.

We thank the review for bringing these to our attention. We have clarified these in the text for SM and MM:

“...including introducing single mismatches (SM) or multiple mismatches (MM) into the guide sequence at different locations”

And for RSR:

“Similar to the prior work¹⁵, applying the transformation assay using minimal CRISPR arrays containing a spacer flanked by full-length repeats (RSR)....”

And SR:

“We also expressed a semi-processed crRNA containing a 20-bp spacer followed by a full-length repeat (SR)....”

6. Line 96: Why do the crRNA sizes vary. Is this due to improper processing?

We believe the variations can be attributed to incomplete processing--something we have seen with SpCas9 in *E. coli* (PMID = 35314780)--as well as the use of two different promoters (P_n, P_s) that may have different transcriptional start sites. To capture these assertions, we added the following to p. 6:

“The crRNA sizes also varied, possibly due to incomplete crRNA processing in *E. coli* as well as differences in transcriptional start sites between promoters; the size variations could be contributing to the extent of DNA targeting.”

7. Line 140, 142: Please indicate Cas9 here was SpyCas9 to be consistent with fig 2C labels.

We updated the text to say SpyCas9 rather than Cas9 to more clearly indicate which variant we were referring to and to match the figures.

8. Line 315: please include concentrations of the antibiotics used in all instances, either in the methods or supplementary table.

We included a new section in the methods to clarify the antibiotics and final concentrations used for each strain.

9. Line 317: what was the back-dilution ratio? Add this detail throughout the methods

We have updated the methods to include the back-dilution ratio that was used.

10. Fig 1C, consider including labels (RSR, SR, sg) to the figures on top of the blot

We have removed the cartoons above the blots in Figs. 1C and S1C and replaced the cartoons with the sample information with Ps RSR, Pn RSR, etc. to simplify the figures.

11. Fig 1B, lower panel – is there a reason kefC is missing from Ps RSR?

We screened a collection of genomic sites with Ps sgRNA (see Fig. S1) but did not test all of them with the different gRNA formats. It was not intentionally left out.

12. Fig1B, it may be worth splitting this into two panels 1B for top graphs and 1C for bottom graphs and linking to paragraph in line 79 to 88.

We considered the reviewer's suggestion, although we decided to keep the panels in Fig. 1B grouped together since they all feature variations on the crRNA format.

13. Fig S1A, add figure legend to indicate these data are WT only.

We thank the reviewer for pointing out this oversight. We have updated all of the figures in the main text and SI to include the legend with the strain information (WT vs $\Delta recA$).

14. Fig 2B, consider adding NT, T labels. And in other figure panels where they are missing.

We thank the reviewer for bringing this oversight to our attention. We have updated the figures to include a legend in each panel with a graph to indicate the strain and/or NT, T labels (in the case where it isn't clear in on the x-axis (e.g. Figs. 2B, 2C).

15. Fig 5B, are the figures for MM2 and MM3 swapped?

We can confirm that the figures are not swapped. We assigned labels to the constructs in sequential order, where the number does not reflect the number of mismatches in the guide. We have updated the text to clarify this.

16. Line 344: was sanger sequencing performed only for the 2 nt deletion experiments? Is there a possibility of improper HR that can introduce SNPs in the HA?

When assessing the 2-nt deletion, the Sanger sequencing results were further used to align the sequence with the reference genome. No SNPs were detected within the HA region. We have updated the methods section accordingly. While we did not observe SNPs in that experiment, we

cannot rule out the possibility of improper HR in the other contents. However, this would be surprising given the precise nature of HR.

a. line 259: “precise genome editing”: unless the edited samples were also sequenced to confirm then nature of RT insertion, it may be inaccurate to use the term “precise”. Colony PCRs performed in this study only show the presence of the edit at the intended target site.

We consider a “precise” edit as one in which the intended edit was introduced. That contrasts with a “random” or “unintended” edit often appearing as an indel. This should follow how the field generally defines a precise edit, which is unrelated to the extent of additional editing well outside of the target site.

17. Line 381: “back-diluted in LB medium with cm and”. what is “cm”?

Cm is chloramphenicol. We have updated the methods section to clarify this.

18. Line 338: Please include agarose gel conditions throughout methods as done in lines 375, 393

We have updated the methods section to more clearly outline the agarose gel conditions used. For example, we have added the following text to the methods:

“The digested products were resolved on a 1.5% agarose gel (or 1% for the Cas12a experiments) in Tris-acetate EDTA (TAE) buffer run at 80V for 40 minutes and subsequently stained in ethidium bromide. “

Reviewer #2 (Remarks to the Author):

Summary

In this manuscript, the authors describe an approach to improve CRISPR-driven genome editing in bacteria by systematically attenuating DNA targeting activity. Previous methods require DNA recombinases and Cas nucleases in tandem and are often characterized by low transformation and editing efficiencies. Here, the authors found that perturbing CRISPR activity, typically by modifying gRNA structure or expression levels, can increase colony counts and recombination efficiency in CRISPR-driven editing. These improvements removed the need for exogenous recombinases in *E. coli* and enabled editing in several non-model bacterial strains. The new approaches described in this manuscript will be of broad interest to the bacterial gene editing and synthetic biology communities, and are well-suited for publication after the authors consider the comments below.

We thank the reviewer for their concise and supportive summary. The comments below also helped us better clarify the proposed model and general decisions made throughout the work as well as how to make the technology more accessible to the community at large.

Major Comments

1. On line 259, the authors describe a general model to explain their results: “This seemingly paradoxical concept--making DNA targeting weaker can improve editing--appears to emerge from the cells having time to repair cut genomic DNA with other copies of the chromosome or a provided RT.” Can the authors elaborate and discuss this point further? Once a cut happens, presumably the cell needs to repair it before replication, otherwise it dies. How does attenuating DNA targeting affect the time window between cleavage and replication? Or is the idea that rapidly dividing cells have multiple copies of the genome and efficient cutting will cleave all copies so the cells die if repair does not occur, but attenuated cutting means some cells will survive and the population will have multiple attempts through multiple cell cycles to introduce the edit? Or could the observed effects arise if the attenuated CRISPR complexes more readily dissociate from DNA after cleavage, allowing the repair machinery easier access?

We agree with the reviewer that our proposed model could be more extensively described, and the reviewer raises multiple valid reasons to explain our observed phenomenon. We therefore expanded our description of the general model with the following on p. 13 of the discussion:

“This seemingly paradoxical concept--making DNA targeting weaker can improve editing--may be explained by the cells having time to repair cut genomic DNA with other copies of the chromosome or a provided RT. During exponential growth, bacteria initiate genome replication multiple times in one cell cycle to keep pace with cell division. Here, attenuated targeting would leave more genomic copies intact and thus available for templated repair. However, other mechanisms could be at work. For instance, some atgRNA modifications

could promote active release of cut DNA, allowing faster access by the repair machinery. In addition, attenuated targeting may lead to nicking rather than cleavage of the DNA target^{28,51}, which can also be repaired through homologous recombination⁵².”

2. Can the authors address how this editing system would work in practice for engineering edited strains? Specifically, would plasmid curing be used to remove the CRISPR system components and/or would non-replicating plasmids be effective? Additional discussion would be sufficient, no new experiments are needed.

We thank the reviewer for this suggestion, and we agree some discussion of how our approach would be practically implemented would be helpful. We therefore added the following to p. 15 of the discussion:

“Once an appropriate mode of attenuated targeting is identified, editing would follow a series of steps paralleling current use of traditional CRISPR-based editing techniques in bacteria. First, the designed atgRNA and repair template would be cloned into a plasmid construct—ideally with the CRISPR nuclease to generate an all-in-one plasmid. To ensure all constructs are removed from the edited strain for downstream use, the plasmid could be encoded with an origin-of-replication that is temperature-sensitive or can be easily cured. Non-replicating repair templates such as an oligonucleotide, linear DNA, or a non-replicating plasmid could also be used, although the editing efficiency with attenuated targeting will likely be reduced because the repair template would not be maintained. A recombinase system, such as λ -red, can also be introduced to further improve or even achieve editing, as we observed for large deletions in *K. oxytoca*.”

We note that guidance on selecting an appropriate mode of attenuated targeting is now described in the prior paragraph in the discussion.

Minor comments

1. In the introduction line 30 it took a few reads to parse and understand this sentence: “Despite the paradigm of utilizing chromosomal cleavage to counterselect against unedited cells, prior work reported an intriguing exception: repair of chromosomal cleavage by Cas9 in *Escherichia coli* through endogenous homologous recombination.” In contrast, the same point on line 54 was very clear and immediately understandable: “We were initially intrigued why targeting some locations in the *E. coli* genome with the *Streptococcus pyogenes* Cas9 led to RecA-dependent homologous recombination rather than cell death.” The authors might consider rephrasing the statement on line 30.

We agree this sentence can be simplified, especially as part of the introduction. That section now reads the following on p. 3:

“Prior work reported an intriguing exception to CRISPR-based counterselection: chromosomal cleavage by Cas9 in *Escherichia coli* can be actively repaired through homologous recombination¹⁵.”

2. The authors used Cas9 for *K. oxytoca* and Cas12a in *K. pneumoniae*. Is there a reason different Cas systems were used and are there general considerations that guided this decision?

The reasons are mostly historical: the Strowig lab had been actively using the existing Cas9 system in *K. oxytoca*, while the Beisel lab had already started using Cas12a in *K. pneumoniae*. For this work, these differences presented the opportunity to test an existing editing system in one example strain and a new system in another example strain. To better clarify this rationale, we added the following to p. 12:

“We utilized Cas12a to explore CRISPR-driven editing outside of *E. coli* utilizing a nuclease besides Cas9.”

3. The labels on Figure 1C are inconsistent with the labels in Fig 1B&D, making it difficult to compare between panels. It would also be helpful if the authors could identify the specific bands corresponding to each expression construct. Can any of the higher molecular weight bands be identified?

We agree with the reviewer that the labeling scheme could be more straightforward. Therefore, we updated the labeling in Fig. 1C and S1C-D to include the label names “Pn RSR, Ps RSR, etc” instead of the cartoon. We have put labels on the right hand side of the blots to indicate which bands should correspond to the mature sgRNA and mature crRNAs. The higher molecular weight bands are expected to represent incomplete processing products (doi: 10.1038/s41564-022-01074-3).

4. On line 93, the authors write: “Northern blotting analysis on the complete set of gRNAs targeting lacZ1 revealed an inverse correlation between the abundance of the final gRNA (Ps sgRNA > Ps SR > Pn SRS ≈ Pn SR > Ps RSR) with colony counts in the presence of recA.” Can the authors provide a quantitative analysis based on band intensities in the Northern blot?

As suggested by the reviewer, we quantified band intensities and compared these values to the colony counts for each construct. The resulting plots shown below and incorporated as Fig. S1B showed a general correlation between band intensities and enhanced survival with RecA, supporting the impact of gRNA expression levels on cell death.

We also updated the related text on p. 6 to read the following:

“In line with our expectation, northern blotting analysis on the complete set of gRNAs targeting *lacZ1* revealed that lower final gRNA abundances were tied to improved survival in the presence of *recA* (Figs. 1C and S1B).”

5. On line 187, the authors write: “High colony counts and editing was lost in the absence of *recA* (Fig. S2F), confirming the importance of RecA-mediated homologous recombination with atgRNAs.” Fig S2F is an important control that clearly supports the model that RecA-mediated repair is responsible for recombination. The authors could consider moving this point to a main text figure panel.

We agree that this is an important conclusion to support that editing is occurring through the RecA-mediated pathway. Therefore we moved that panel into the main text figure and now is Fig. 3D.

6. Several figure panels show a limit of detection for colony count assays, and this limit of detection appears to vary between figures. Can the authors clarify in the methods section how the limit of detection was calculated?

The limit-of-detection depends on the volume of cells plated for each experiment. For editing experiments, three to four technical replicates of the dilution series were plated and therefore this reduced the limit of detection. For genome targeting assays, one to two technical replicates were plated and therefore have a higher limit-of-detection. We have updated the methods section to include this information.

7. On line 215, the authors note that lambda red recombination was still necessary for editing in *K. oxytoca*. It appears that lambda red was not necessary for editing in *K. pneumoniae*. Can the authors provide any guidance or predictions on when lambda red will be necessary? Would users need to experimentally test each new strain, or are there differences in endogenous recombinases that might be predictive?

Based on our current information, we can't decidedly explain why lambda-RED was required in *K. oxytoca* but not in *K. pneumoniae*. At most, we can recommend testing a recombinase system if it is available. To address the requirement for lambda-RED in *K. oxytoca*, we added the following to p. 11:

"The necessity of λ -red deviated from what we observed in *E. coli*, although this may be attributed to the different strain, the use of a linear repair template, or the larger deletion at this particular genomic site."

We also added the following to p. 15 to provide some guidance:

"A recombinase system, such as λ -red, can also be introduced to further improve or even achieve editing, as we observed for large deletions in *K. oxytoca*."

8. On line 220, the authors write: "For the sp47 sgRNA, the atgRNA boosted colony counts to be consistently above the limit of detection and yielded an editing frequency of ~83% (Fig. 4B)." In the same figure, several unmodified sgRNAs appear to show relatively high editing efficiencies and colony counts, and the unmodified sp55 sgRNA appears to give the best combination of colony counts and recombination efficiency. Can the authors comment on this observation? Do the authors expect that if enough target sites are screened, suitable sgRNAs might be identified with high editing efficiency for other editing targets? Could the sp55 sgRNA be a potentially poor target site or misfolded sgRNA that is already partially attenuated?

The reviewer makes an astute observation. To better capture the impact of sg55 on colony counts and editing frequencies as well as possible underlying reasons, we added the following to p. 12:

"sp55 was particularly intriguing given the higher colony counts (~104 transformants) and editing frequencies (~60%) without any modification (Fig. 4B), suggesting that the sgRNA already exhibited attenuated targeting possibly through sgRNA misfolding or poor target recognition."

We also added the following to the discussion on p. 15 to generalize this observation:

"It is also possible that an unmodified sgRNA in itself exhibits poor targeting activity and thus would promote CRISPR-driven editing."

Reviewer #3 (Remarks to the Author):

In this work, the authors demonstrated that systematically attenuating DNA targeting activity can achieve CRISPR-driven editing in bacteria, greatly boosting colony counts and even increasing the frequency of precise genome editing. However, the concept of attenuating Cas protein expression of gRNA expression/format is not new. The authors do not provide novel concept or design, nor do they demonstrate very solid data to prove its suitability to publish in Nature Communications. I would recommend rejection of this paper.

We thank the reviewer for taking the time to consider our work. The reviewer raises a few points worth addressing here.

One point was that attenuating DNA targeting is not a new concept. We disagree with this assertion and hold that the idea of restraining targeting activity represents an unexplored concept—let alone one that can improve CRISPR-based editing in bacteria. At most, others had explored the impact of mutating the guide or repeat, tuning component expression, or modifying the Cas nuclease. However, in each example, the goals were entirely unrelated to our work (e.g., creating orthogonal repeats, reducing off-targeting, appending functional sequences to the gRNA, generating high-fidelity nucleases). Our distinct idea was utilizing the “undesirable” alterations from each example to create a new paradigm for CRISPR-based editing in bacteria. By following this idea, we synthesized otherwise unrelated areas of CRISPR technologies (e.g., target mismatches, crRNA biogenesis, sgRNA modifications) to develop a general framework for attenuating DNA targeting. We believe this innovative approach will invite follow-on work to devise the best means of attenuating DNA targeting at different sites across bacteria and one that introduces a distinct concept within CRISPR technologies: that “bad” targeting can yield “good” editing.

Another point was that we did not demonstrate solid data supporting the concept. Contrasting with this assertion, our work included the following:

- Evaluated eight different means of achieving attenuated targeting: introduced crRNA processing, tuned gRNA expression, high-fidelity Cas nucleases, targets flanked by non-canonical PAMs, guide:target mismatches, repeat mutations, external hairpins appended to the guide, and internal hairpins appended to the guide)
- Utilized Cas9 and Cas12a nucleases
- Tested small and large edits
- Applied our approach to three bacterial species (*Escherichia coli*, *Klebsiella oxytoca*, *Klebsiella pneumoniae*)
- Directly compared our approach to standard CRISPR-based editing
- Provided a proof-of-principle demonstration by creating a new resistance marker for use in *Klebsiella pneumoniae*

All experiments were also performed with biological replicates and appropriate controls. Given these efforts, we believe that we thoroughly demonstrated the concept of attenuating DNA targeting to achieve CRISPR-driven editing in bacteria.

Other major critiques:

1. The manuscript is not well written in terms of clarity. The authors claim the use of attenuated gRNA. However, it is unclear what this is after reading the abstract and introduction.

In response to this comment, we have critically reviewed the manuscript and improved any instances in which clarity was lacking. This included any instances noted by the other two reviewers.

We also revisited how attenuated gRNAs are defined. The abstract should be sufficient, as “attenuated” by definition means less active. However, attenuated gRNAs could be more explicitly described in the introduction. We therefore added the following to p. 4:

“The most tunable approach, which involved what we call attenuated gRNAs (atgRNAs) named based on modifications designed to interfere with gRNA function,”

2. Fig. 1, the effects of Rec A on recombination in *E. coli* is well known and has been studied in CRISPR previously.

We fully agree with the reviewer that RecA is well established as a central player in homologous recombination in bacteria including *E. coli*. However, the intent of Figure 1 was not to reveal new functions of RecA in *E. coli*; the intent was to show that the apparent cytotoxicity of genome targeting with Cas9 was influenced by how the genome-targeting gRNA is encoded and expressed. The *recA*-deletion was included to demonstrate that higher colony counts were through RecA-mediated recombination, in line with the cells actively repairing genomic cleavage by Cas9. This insight was the basis to establish attenuated DNA targeting as a means to increase colony counts and even improve the efficiency of genome editing in bacteria.

3. The authors claim that this method obviates the need of recombinase. However, the use of recombinase or lambda red provides excellent genome editing efficiency and low off-target effects. The CRISPR-lambda red system also enable multiplexing and rational metabolic engineering of *E. coli*. The authors only provide some conceptual data without demonstrating its advantage over other approaches. It's unlikely that this approach will be used broadly.

We agree with the reviewer that recombinases and recombinase systems such as λ -red have proven invaluable tools for genome editing in bacteria. At the same time, they must be heterologously expressed, requiring an extra component that complicates genome editing efforts. Furthermore, few bacteria possess compatible recombinases or recombination systems, limiting their utility for genome editing. Even when a compatible recombinase or recombination system is available, it's not a cure-all: recombination still occurs at lower and often variable frequencies that decrease exponentially in multiplex. These challenges spurred the introduction of CRISPR-based

counterselection in combination with recombinases, although doing so came with the disadvantages of reduced colony counts and the prevalence of escape mutants. Our approach offered the unique opportunity to shed this extra component without eliminating genome editing.

The reviewer also claimed that there were no demonstrations of comparisons over other approaches. By demonstrating genome editing without a recombinase, we show that the recombinase can be eliminated--an advantage when streamlining the constructs necessary for genome editing. In addition, we also include the original sgRNA for Cas9 or gRNA for Cas12a, representing traditional editing with CRISPR-based counterselection, in almost every editing experiment. This comparison revealed that attenuating DNA targeting could increase colony counts and, in many cases, also improve editing efficiency. We believe these comparisons are sufficient to argue for our approach over existing techniques for genome editing in bacteria.

Reviewers' Comments:

Reviewer #1:

Remarks to the Author:

The authors have taken steps to address my comments with the revised version of the manuscript.
The manuscript can be accepted for publication.

Reviewer #2:

Remarks to the Author:

The authors have satisfactorily and thoroughly addressed the reviewer comments and the manuscript should be published.

Reviewer comments NCOMMS-22-36117A

We thank all reviewers for their consideration, input, and helpful comments throughout the review process. We address each comment below. Accompanying changes in the main text are in red.

Reviewer #1 (Remarks to the Author):

The authors have taken steps to address my comments with the revised version of the manuscript. The manuscript can be accepted for publication.

We thank the reviewer for their feedback and support.

Reviewer #2 (Remarks to the Author):

The authors have satisfactorily and thoroughly addressed the reviewer comments and the manuscript should be published.

We thank the reviewer for their feedback and support.